# Fine-Tuning Language Models with Just Forward Passes

**Sadhika Malladi**[*]     **Tianyu Gao**[*]     **Eshaan Nichani**     **Alex Damian**

**Jason D. Lee**          **Danqi Chen**          **Sanjeev Arora**

Princeton University
{smalladi, tianyug, eshnich, ad27, jasonlee, danqic, arora}@princeton.edu

## Abstract

Fine-tuning language models (LMs) has yielded success on diverse downstream tasks, but as LMs grow in size, backpropagation requires a prohibitively large amount of memory. Zeroth-order (ZO) methods can in principle estimate gradients using only two forward passes but are theorized to be catastrophically slow for optimizing large models. In this work, we propose a memory-efficient zeroth-order optimizer (**MeZO**), adapting the classical ZO-SGD method to operate in-place, thereby fine-tuning LMs with *the same memory footprint as inference*. For example, with a single A100 80GB GPU, MeZO can train a 30-billion parameter model, whereas fine-tuning with backpropagation can train only a 2.7B LM with the same budget. We conduct comprehensive experiments across model types (masked and autoregressive LMs), model scales (up to 66B), and downstream tasks (classification, multiple-choice, and generation). Our results demonstrate that (1) MeZO significantly outperforms in-context learning and linear probing; (2) MeZO achieves comparable performance to fine-tuning with backpropagation across multiple tasks, with up to $12\times$ memory reduction and up to $2\times$ GPU-hour reduction in our implementation; (3) MeZO is compatible with both full-parameter and parameter-efficient tuning techniques such as LoRA and prefix tuning; (4) MeZO can effectively optimize non-differentiable objectives (e.g., maximizing accuracy or F1). We support our empirical findings with theoretical insights, highlighting how adequate pre-training and task prompts enable MeZO to fine-tune huge models, despite classical ZO analyses suggesting otherwise.[1]

## 1   Introduction

Fine-tuning pre-trained language models (LMs) has been the dominant methodology for solving many language tasks [28], adapting to specialized domains [42], or incorporating human instructions and preferences [73]. However, as LMs are scaled up [13, 72], computing gradients for backpropagation requires a prohibitive amount of memory – in our test, up to $12\times$ the memory required for inference – because it needs to cache activations during the forward pass, gradients during the backward pass, and, in the case of Adam [52], also store gradient history (see Section 3.4 for a detailed analysis).

As a result, while it is possible to run inference with a 30-billion (30B) parameter LM on a single Nvidia A100 GPU (with 80GB memory), backpropagation with Adam is feasible only for a 2.7B LM. Parameter-efficient fine-tuning methods (PEFT [46, 57, 54]) update just a fraction of the network

---

[*]Equal contribution and corresponding authors.
[1]Our code is available at https://github.com/princeton-nlp/MeZO.

37th Conference on Neural Information Processing Systems (NeurIPS 2023).

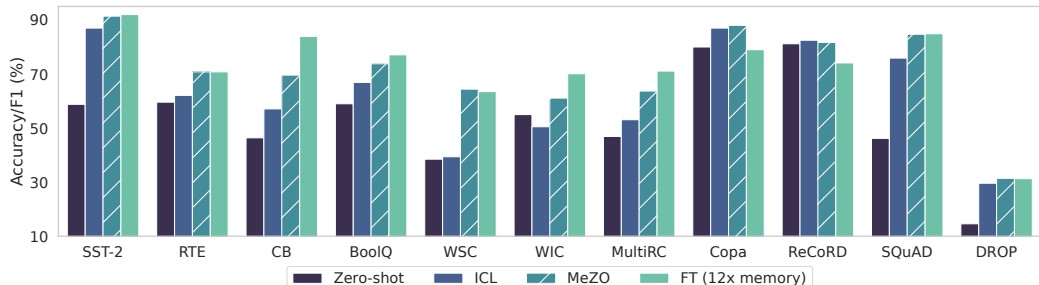

Figure 1: OPT-13B results with zero-shot, in-context learning (ICL), MeZO (we report the best among MeZO/MeZO (LoRA)/MeZO (prefix)), and fine-tuning with Adam (FT). MeZO demonstrates superior results over zero-shot and ICL and performs on par with FT (within 1%) on 7 out of 11 tasks, despite using only 1/12 memory. See Table 1 for detailed numbers and Figure 3 for memory profiling.

parameters, but still need to cache many activations, because the tuned parameters are scattered throughout the model. In our tests, fine-tuning an OPT-13B model with full parameter tuning or PEFT requires $12\times$ and $6\times$ more memory than inference respectively.

*In-context learning* (ICL [13]) has allowed solving many tasks with a single inference pass, during which the model processes labeled examples (*demonstrations*) in its context and then outputs a prediction on a test example. While this allows for quick adaptation of the model to specific use cases, current models allow a limited context size (and thus, limited demonstrations) and the performance is sensitive to the formatting and choice of demonstrations [60, 66]. ICL can slow with the number of demonstrations, and it often performs worse than fine-tuning of medium-sized models [13].

Backpropagation also cannot optimize non-differentiable criteria, which have gained popularity in fine-tuning LMs according to human preference scores or set safety standards [89, 73]. Typically, these adaptations involve expensive reinforcement learning from human feedback (RLHF [20]).

A classical zeroth-order optimization method, ZO-SGD [88], uses only differences of loss values to estimate the gradients. Thus, in principle, the method can update neural networks with just forward passes, though naive implementation still doubles the memory overhead and classical lower bounds [69, 32] suggest that convergence slows linearly with model size. As such, ZO methods have been applied in deep learning settings to find adversarial examples or tune input embeddings [91, 90] but not to directly optimize large-scale models (see Liu et al. [61] for a survey).

In this work, we propose a memory-efficient zeroth-order optimizer (MeZO), which adapts the classical ZO-SGD algorithm and reduces its memory consumption *to the same as inference*. We apply MeZO to fine-tune large LMs and show that, both empirically and theoretically, MeZO can successfully optimize LMs with billions of parameters. Specifically, our contributions are:

1. In MeZO, we adapt the ZO-SGD algorithm [88] and a number of variants to operate in-place on arbitrarily large models with almost no memory overhead (see Algorithm 1 and Section 2).

2. We conduct comprehensive experiments across model types (masked LM and autoregressive LM), model scales (from 350M to 66B), and downstream tasks (classification, multiple-choice, and generation). MeZO consistently outperforms zero-shot, ICL, and linear probing. Moreover, with RoBERTa-large, MeZO achieves performance close to standard fine-tuning within 5% gap; with OPT-13B, MeZO outperforms or performs comparably to fine-tuning on 7 out of 11 tasks, despite requiring roughly $12\times$ less memory (Figure 1 and Section 3). In our implementation, MeZO requires only half as many GPU-hours as Adam fine-tuning for a 30B model (see Appendix F.6).

3. We demonstrate MeZO's compatibility with full-parameter tuning and PEFT (e.g., LoRA [46] and prefix-tuning [57]) in Section 3.

4. Further exploration showcases that MeZO can optimize non-differentiable objectives such as accuracy or F1 score, while still requiring only the same memory as inference (Section 3.3).

5. Our theory suggests that adequate pre-training ensures the per-step optimization rate (Theorem 1) and global convergence rate (Lemma 3) of MeZO depend on a certain condition number of the landscape (i.e., the local effective rank, see Assumption 1) instead of numbers of parameters. This

---

**Algorithm 1:** MeZO

---

**Require**: parameters $\boldsymbol{\theta} \in \mathbb{R}^d$, loss $\mathcal{L} : \mathbb{R}^d \to \mathbb{R}$, step budget $T$, perturbation scale $\epsilon$, batch size $B$, learning rate schedule $\{\eta_t\}$

---

**for** $t = 1, ..., T$ **do**
    Sample batch $\mathcal{B} \subset \mathcal{D}$ and random seed $s$
    $\boldsymbol{\theta} \leftarrow$ `PerturbParameters`$(\boldsymbol{\theta}, \epsilon, s)$
    $\ell_+ \leftarrow \mathcal{L}(\boldsymbol{\theta}; \mathcal{B})$
    $\boldsymbol{\theta} \leftarrow$ `PerturbParameters`$(\boldsymbol{\theta}, -2\epsilon, s)$
    $\ell_- \leftarrow \mathcal{L}(\boldsymbol{\theta}; \mathcal{B})$
    $\boldsymbol{\theta} \leftarrow$ `PerturbParameters`$(\boldsymbol{\theta}, \epsilon, s)$         ▷ `Reset parameters before descent`

    `projected_grad` $\leftarrow (\ell_+ - \ell_-)/(2\epsilon)$
    Reset random number generator with seed $s$         ▷ `For sampling` $z$
    **for** $\theta_i \in \boldsymbol{\theta}$ **do**
        $z \sim \mathcal{N}(0, 1)$
        $\theta_i \leftarrow \theta_i - \eta_t * $ `projected_grad` $* z$
    **end**
**end**

**Subroutine** `PerturbParameters`$(\boldsymbol{\theta}, \epsilon, s)$
    Reset random number generator with seed $s$         ▷ `For sampling` $z$
    **for** $\theta_i \in \boldsymbol{\theta}$ **do**
        $z \sim \mathcal{N}(0, 1)$
        $\theta_i \leftarrow \theta_i + \epsilon z$         ▷ `Modify parameters in place`
    **end**
    **return** $\boldsymbol{\theta}$

---

result is in sharp contrast to existing ZO lower bounds [69, 32] suggesting that the convergence rate can slow proportionally to the number of parameters (Section 4).

## 2 Zeroth-order optimization

Zeroth-order (ZO) optimizers have long been studied in the context of convex and strongly convex objectives. In the following, we first introduce a classical ZO gradient estimator, SPSA (Definition 1 [88]) and the corresponding SGD algorithm, ZO-SGD (Definition 2). Then we describe MeZO, our in-place implementation that requires the same memory as inference in Section 2.1 and Algorithm 1. We highlight that SPSA can also be used in more complex optimizers, such as Adam, and we provide memory-efficient implementations for those algorithms too (Section 2.2).

Consider a labelled dataset $\mathcal{D} = \{(\boldsymbol{x}_i, \boldsymbol{y}_i)\}_{i \in [|\mathcal{D}|]}$ and a minibatch $\mathcal{B} \subset \mathcal{D}$ of size $B$, we let $\mathcal{L}(\boldsymbol{\theta}; \mathcal{B})$ denote the loss on the minibatch. We introduce a classical ZO gradient estimate in this setting.[2]

**Definition 1** (Simultaneous Perturbation Stochastic Approximation or SPSA [88])**.** *Given a model with parameters $\boldsymbol{\theta} \in \mathbb{R}^d$ and a loss function $\mathcal{L}$, SPSA estimates the gradient on a minibatch $\mathcal{B}$ as*

$$\widehat{\nabla}\mathcal{L}(\boldsymbol{\theta}; \mathcal{B}) = \frac{\mathcal{L}(\boldsymbol{\theta} + \epsilon\boldsymbol{z}; \mathcal{B}) - \mathcal{L}(\boldsymbol{\theta} - \epsilon\boldsymbol{z}; \mathcal{B})}{2\epsilon}\boldsymbol{z} \approx \boldsymbol{z}\boldsymbol{z}^\top \nabla\mathcal{L}(\boldsymbol{\theta}; \mathcal{B}) \tag{1}$$

*where $\boldsymbol{z} \in \mathbb{R}^d$ with $\boldsymbol{z} \sim \mathcal{N}(0, \boldsymbol{I}_d)$ and $\epsilon$ is the* perturbation scale. *The $n$-SPSA gradient estimate averages $\widehat{\nabla}\mathcal{L}(\boldsymbol{\theta}; \mathcal{B})$ over $n$ randomly sampled $\boldsymbol{z}$.*

SPSA requires only *two forward passes* through the model to compute the gradient estimate (for $n$-SPSA, each estimate requires $2n$ forward passes). As $\epsilon \to 0$, the SPSA estimate can be understood as a rank-1 reconstruction of the gradient. During training, $n$ can be treated as a hyperparameter and follow a schedule [11, 15], though in cursory experiments (Appendix A), $n = 1$ is the most efficient.

---

[2]The original SPSA algorithm [88] perturbs the model by $1/\boldsymbol{z}$ and thus requires that $\boldsymbol{z}$ has finite inverse moments, precluding the choice of $\boldsymbol{z}$ as Gaussian. $1/\boldsymbol{z}$ is very large with high probability for a zero-mean Gaussian $\boldsymbol{z}$, so we adopt the standard in many theoretical [70, 32] and empirical [64] works and perturb the parameters by $\boldsymbol{z}$ with $\boldsymbol{z}$ as a Gaussian random variable.

We use $n = 1$ as the default. It is widely known that the SPSA estimate can be used to replace the backpropagation gradient in any optimizer such as SGD.

**Definition 2** (ZO-SGD). *ZO-SGD is an optimizer with learning rate $\eta$ that updates parameters as $\boldsymbol{\theta}_{t+1} = \boldsymbol{\theta}_t - \eta\widehat{\nabla}\mathcal{L}(\boldsymbol{\theta}; \mathcal{B}_t)$ where $\mathcal{B}_t$ is the minibatch at time $t$ and $\widehat{\nabla}\mathcal{L}$ is the SPSA gradient estimate.*

### 2.1 Memory-efficient ZO-SGD (MeZO)

The vanilla ZO-SGD algorithm costs twice the memory of inference, as it needs to store $\boldsymbol{z} \in \mathbb{R}^d$. We propose a memory-efficient implementation of ZO-SGD called **MeZO**, as illustrated in Algorithm 1. At each step, we first sample a random seed $s$, and then for each of $\boldsymbol{z}$'s four uses in Algorithm 1, we reset the random number generator by $s$ and *resample* the relevant entry of $\boldsymbol{z}$. Using this in-place implementation, MeZO has a memory footprint equivalent to the inference memory cost.

We note that Algorithm 1 describes perturbing each parameter separately, which may be time-consuming for large models. In practice, we can save time by perturbing an entire weight matrix instead of each scalar independently. This incurs an additional memory cost as large as the largest weight matrix; usually, this is the word embedding matrix (e.g., 0.86GB for OPT-66B).

**Storage Efficiency of MeZO.** Parameter-efficient fine-tuning (PEFT) techniques fine-tune just a fraction of the network parameters and have thus been proposed as a way to reduce the storage costs of fine-tuned model checkpoints. Fine-tuning with MeZO reduces the storage cost of the resulting checkpoint far more than popular PEFT techniques (e.g., LoRA [46] and prefix tuning [57]). We reconstruct the MeZO trajectory using a single seed, which spawns step-wise seeds to sample $\boldsymbol{z}$, and the `projected_grad` at each step.[3] As such, for fine-tuning a 66B model, MeZO requires saving the seed plus 20,000 (steps) $\times$ 2 bytes, which is less than 0.1MB. LoRA fine-tunes 19M parameters and requires 38MB storage, and prefix tuning fine-tunes 6M parameters and requires 12MB storage.

### 2.2 MeZO extensions

We note that SPSA is a popular ZO gradient estimator but not the only one. Many one-point gradient estimators have been proposed in past works [34, 87, 95], and using such estimators in place of SPSA would halve the training time. However, cursory experiments with one such promising estimator [113] reveal that these are not as efficient as SPSA when fixing the number of forward passes (Appendix B.5). As such, we implement MeZO with the SPSA estimator.

MeZO can also be combined with other gradient-based optimizers, including SGD with momentum or Adam. Though naive implementation would require additional memory to store the gradient moment estimates, MeZO-momentum and MeZO-Adam alleviate such overhead by recomputing the moving average of the gradients using saved past losses and $\boldsymbol{z}$ (see Appendix B for a full discussion).

We also note that all of the coordinates of the SPSA gradient estimate have the same scale, but deep Transformers can have gradients of different scales for each layer [59, 61]. As such, we draw inspiration from layerwise adaptive optimizers [109, 110] to design several MeZO variants. Cursory experiments showed that these algorithms are not more efficient (in terms of forward passes), but we nevertheless present them as potential optimizers for more complex objectives. See Appendix B.

**Forward Auto-Differentiation** Note that $\boldsymbol{z}^\top\nabla\mathcal{L}(\boldsymbol{\theta}; \mathcal{B})$ is a Jacobian-vector product (JVP), which can be computed in parallel with an inference pass with excess memory consumption equivalent to that of the largest activation in the network [40]. In this case, $\boldsymbol{z}$ must be stored on the GPU in order to construct the gradient estimate, so this procedure requires slightly more than two times the memory needed for inference. We analyze this algorithm in detail in Appendix D. Note that using a non-zero $\epsilon$ in SPSA, which is not possible through the JVP method, may boost generalization by promoting a sharpness-minimizing term. Past works (e.g., Baydin et al. [9]) have also studied JVP-based training but achieved limited empirical success.

---

[3]Note that this reconstruction requires no additional forward passes through the model and no access to the data used during fine-tuning, since `projected_grad` implicitly encodes this information.

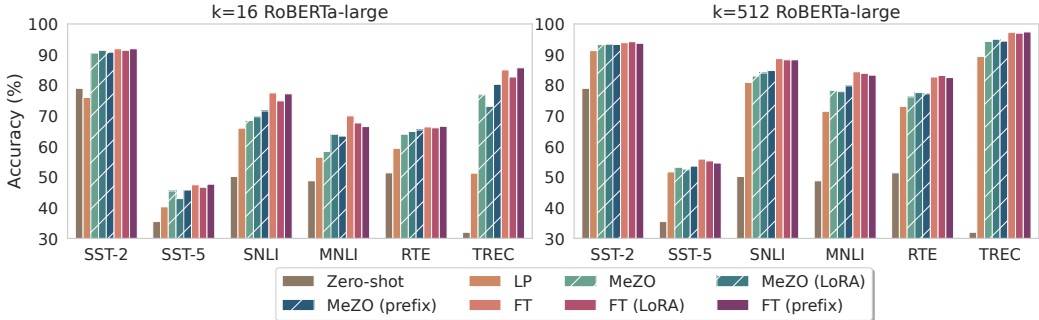

Figure 2: Experiments on RoBERTa-large. We report zero-shot, linear probing (LP), and MeZO and fine-tuning (FT) with full parameter, LoRA, and prefix-tuning. MeZO outperforms zero-shot and LP and approaches FT (within 5% for $k = 512$) with much less memory. Detailed numbers in Table 18.

## 3 Experiments

Preliminary experiments (Appendix A) show that MeZO only works when using prompts [13, 84, 35]. Past works [83, 67] have demonstrated how the inclusion of a suitable prompt ensures the fine-tuning objective is closely related to the pre-training one. In Section 4, we extend these ideas to show how using a simple prompt simplifies the fine-tuning optimization procedure, thereby enabling zeroth order methods to work efficiently. All experiments below use prompts detailed in Appendix E.2. All fine-tuning with backpropagation (FT) experiments follow convention and use Adam, though we also report results when performing FT with SGD in Appendix F.

We conduct comprehensive experiments on both medium-sized masked LMs (RoBERTa-large, 350M [65]) and large autoregressive LMs (OPT-13B, 30B, 66B [112]) in few-shot and many-shot settings with prompts. We also explore both full-parameter tuning and PEFT including LoRA [46] and prefix-tuning [57] (see Appendix E.5 for details). We compare MeZO with zero-shot, in-context learning (ICL), linear-probing (LP), and fine-tuning with Adam (FT). MeZO uses substantially less memory than FT but requires significantly more training steps.

We first show that MeZO improves substantially over zero-shot, ICL, and LP across model types, sizes, and task types. Moreover, MeZO performs comparably to FT over a number of tasks, while drastically reducing the memory cost by, for example, 12× on OPT-13B. Further experiments demonstrate that MeZO can optimize non-differentiable objectives, such as accuracy and F1 score (Section 3.3). We compare the memory consumption of ICL, FT, LP, and MeZO in Figures 3 and 4.

### 3.1 Medium-sized masked language models

We conduct experiments with RoBERTa-large on sentiment classification, natural language inference, and topic classification tasks. We follow past works [35, 67] in studying the few-shot and many-shot settings, sampling $k$ examples per class for $k = 16$ and $k = 512$ (details in Appendix E). We run MeZO for 100K steps and fine-tuning for 1000 steps, noting that one MeZO step is substantially faster than one fine-tuning step (see Appendix F.6 for a comparison). We summarize the results from Figure 2 and Table 18 below.

**MeZO works significantly better than zero-shot, linear probing, and other memory-equivalent methods.** On all six diverse tasks, MeZO can optimize the pre-trained model and consistently perform better than zero-shot and linear probing. We also show for several tasks that MeZO can outperform another ZO algorithm, BBTv2 [90], by up to 11% absolute (Appendix F.4).[4]

**With enough data, MeZO achieves comparable performance (up to 5% gap) to FT.** MeZO achieves close-to-fine-tuning performance on $k = 16$, with some tasks only having 2% gaps. When using $k = 512$ data, the gap between MeZO and FT further reduced to within 5% across all tasks.

**MeZO works well on both full-parameter tuning and PEFT.** Full-parameter tuning (MeZO) and PEFT (MeZO with LoRA and prefix-tuning) achieve comparable performance, while MeZO (prefix)

---

[4]BBTv2 can only train low-dimensional projected prefixes instead of the full model.

| Task
Task type | SST-2 | RTE | CB | BoolQ | WSC | WIC | MultiRC | COPA | ReCoRD | SQuAD | DROP |
|---|---|---|---|---|---|---|---|---|---|---|---|
| | | | — | classification — | | | | – multiple choice – | | — generation — | |
| Zero-shot | 58.8 | 59.6 | 46.4 | 59.0 | 38.5 | 55.0 | 46.9 | 80.0 | 81.2 | 46.2 | 14.6 |
| ICL | 87.0 | 62.1 | 57.1 | 66.9 | 39.4 | 50.5 | 53.1 | 87.0 | **82.5** | 75.9 | 29.6 |
| LP | **93.4** | 68.6 | 67.9 | 59.3 | 63.5 | 60.2 | 63.5 | 55.0 | 27.1 | 3.7 | 11.1 |
| MeZO | 91.4 | 66.1 | 67.9 | 67.6 | 63.5 | **61.1** | 60.1 | **88.0** | 81.7 | **84.7** | 30.9 |
| MeZO (LoRA) | 89.6 | 67.9 | 66.1 | **73.8** | **64.4** | 59.7 | 61.5 | 87.0 | 81.4 | 83.8 | **31.4** |
| MeZO (prefix) | 90.7 | **70.8** | **69.6** | 73.1 | 57.7 | 59.9 | **63.7** | 84.0 | 81.2 | 84.2 | 28.9 |
| FT (12x memory) | 92.0 | 70.8 | 83.9 | 77.1 | 63.5 | 70.1 | 71.1 | 79.0 | 74.1 | 84.9 | 31.3 |

Table 1: Experiments on OPT-13B (with 1000 examples). ICL: in-context learning; LP: linear probing; FT: full fine-tuning with Adam. MeZO outperforms zero-shot, ICL, and LP across the board, and achieves comparable (within 1%) or better performance than FT on 7 out of 11 tasks.

| Task | SST-2 | RTE | BoolQ | WSC | WIC | SQuAD |
|---|---|---|---|---|---|---|
| 30B zero-shot | 56.7 | 52.0 | 39.1 | 38.5 | 50.2 | 46.5 |
| 30B ICL | 81.9 | 66.8 | 66.2 | 56.7 | 51.3 | 78.0 |
| 30B MeZO/MeZO (prefix) | **90.6** | **72.6** | **73.5** | **63.5** | **59.1** | **85.2** |
| 66B zero-shot | 57.5 | **67.2** | 66.8 | 43.3 | 50.6 | 48.1 |
| 66B ICL | 89.3 | 65.3 | 62.8 | 52.9 | 54.9 | 81.3 |
| 66B MeZO/MeZO (prefix) | **93.6** | 66.4 | **73.7** | **63.5** | **58.9** | **85.0** |

Table 2: Experiments on OPT-30B and OPT-66B (with 1000 examples). We report the best of MeZO and MeZO (prefix). See Appendix F.2 for more results. We see that on most tasks MeZO effectively optimizes up to 66B models and outperforms zero-shot and ICL.

sometimes outperforms MeZO. We also show in Appendix F.3 that the three variants converge at similar rates, agreeing with our theory in Section 4, which shows that MeZO converges at a rate independent of the number of parameters being optimized.

We show additional results with more FT and MeZO variants in Appendix F.1. We see that (1) ZO-Adam sometimes outperforms ZO-SGD but is not consistent across tasks; (2) LP and then MeZO, as suggested for fine-tuning [53], can sometimes improve the performance.

## 3.2 Large autoregressive language models

With the promising results from RoBERTa-large, we extend MeZO to the OPT family [112], on a scale of 13B (Table 1), 30B, and 66B (Table 2). We select both SuperGLUE [98] tasks[5] (including classification and multiple-choice) and generation tasks. We randomly sample 1000, 500, and 1000 examples for training, validation, and test, respectively, for each datset. We run MeZO for 20K steps and fine-tuning for 5 epochs, or 625 steps, noting that each step of MeZO is substantially faster than fine-tuning (see Appendix F.6 for a comparison). Please refer to Appendix E for details. Table 1 yields the following observations.

**MeZO outperforms memory-equivalent methods and closely approaches fine-tuning results.** We see that on a 13B-parameter scale, MeZO and its PEFT variants outperform zero-shot, ICL, and LP across almost all tasks. When comparing to FT, which costs 12× more memory (Section 3.4), MeZO achieves comparable (within 1%) or better performance on 7 out of the 11 tasks.

**MeZO exhibits strong performance across classification, multiple-choice, and generation tasks.** We investigate MeZO on generation tasks, which are regarded as more intricate than classification or multiple-choice tasks. We evaluate on two question answering datasets, SQuAD [80] and DROP [31]. We use teacher forcing for training and greedy decoding for inference (details in Appendix E).

Table 1 shows that, on all generation tasks, MeZO outperforms zero-shot, ICL, and LP, and achieves comparable performance to FT. Considering that many applications of fine-tuning LMs – including instruction tuning or domain adaptation – target generation tasks, our results underscore the potential of MeZO as a memory-efficient technique to optimize large LMs for realistic and exciting applications.

---

[5]We also include SST-2, which is a simple sentiment classification task that we use for development.

| Model | RoBERTa-large (350M) | | | | OPT-13B |
| Task | SST-2 | SST-5 | SNLI | TREC | SQuAD |
|---|---|---|---|---|---|
| Zero-shot | 79.0 | 35.5 | 50.2 | 32.0 | 46.2 |
| Cross entropy (FT) | 93.9 | 55.9 | 88.7 | 97.3 | 84.2 |
| Cross entropy (MeZO) | 93.3 | 53.2 | 83.0 | 94.3 | 84.7 |
| Accuracy/F1 (MeZO) | 92.7 | 48.9 | 82.7 | 68.6 | 78.5 |

Table 3: MeZO with non-differentiable objectives. For classification ($k = 512$), we use MeZO with full-parameter and optimize accuracy; for SQuAD (1,000 examples), we use MeZO (prefix) and F1.

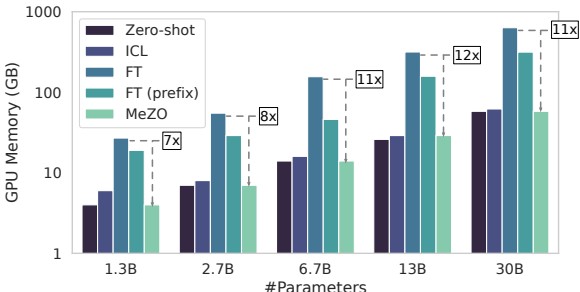

| Hardware | Largest OPT that can fit | | |
|---|---|---|---|
| | FT | FT-prefix | Inference |
| 1×A100 (80GB) | 2.7B | 6.7B | 30B |
| 2×A100 (160GB) | 6.7B | 13B | 66B |
| 4×A100 (320GB) | 13B | 30B | 66B |
| 8×A100 (640GB) | 30B | 66B | 175B$^{†}$ |

Figure 3: GPU memory consumption with different OPT models and tuning methods on MultiRC (400 tokens per example on average).

Figure 4: Largest OPT models that one can tune with specific hardwares and algorithms. † : projected results without actual testing.

**MeZO scales up to 66 billion parameter models.** We demonstrate the efficacy of MeZO on even larger models, up to 66B, in Table 2. While directly fine-tuning models at such scales is extremely costly (Section 3.4), MeZO can effectively optimize these models and outperform zero-shot and ICL.

## 3.3 Training with non-differentiable objectives

We demonstrate the efficacy of MeZO for optimizing non-differentiable objectives through initial experiments. Accuracy and F1 are used as the respective objectives (details in Appendix E.6). Table 3 reveals that MeZO with accuracy/F1 successfully optimizes LMs with superior performance to zero-shot. Although minimizing cross entropy results in stronger performance, these preliminary findings highlight the promising potential of applying MeZO to optimize non-differentiable objectives without clear differentiable surrogates, such as human preferences [73].

## 3.4 Memory usage and wall-clock time analysis

In this section we profile the memory usage of zero-shot, ICL, FT, FT (prefix), and MeZO. We test OPT models of various sizes with Nvidia A100 GPUs (80GB memory) on MultiRC (average #tokens=400), and report the peak GPU memory consumption (details in Appendix E.7).

As shown in Figure 3 (refer to Appendix F.5 for detailed numbers), MeZO exhibits the same memory consumption as zero-shot while offering memory savings of up to 12 times compared to standard FT and 6 times compared to FT (prefix). This advantage enables training larger models within a fixed hardware budget, as illustrated in Figure 4. Specifically, using a single A100 GPU, MeZO allows for tuning a model that is 11 times larger than what is feasible with FT.

In Appendix F.6, we compare the wall-clock time efficiencies of our implementations of MeZO and Adam fine-tuning. MeZO achieves 7.74× per-step speedup and requires 8× fewer GPUs with a 30B model, but takes more steps to converge. Overall, MeZO only requires half as many GPU-hours to fine-tune a 30B model compared to full-parameter fine-tuning. The wall-clock benefit of MeZO is not inherent to the algorithm and is highly dependent on the implementation. We primarily provide this information as a demonstration that MeZO does not take a prohibitively long time to run.

The above measurements are dependent on the computing infrastructure. In Appendix C, we compare the theoretical time-memory tradeoff of MeZO and backpropagation and find that MeZO is always more memory-efficient than backpropagation and is often more time-efficient. The above analyses also do not consider recent advances (e.g., gradient checkpointing [18], FlashAttention [23], and quantization [27]). We leave investigating the how MeZO works with these methods to future work.

# 4   Theory

Our theoretical analysis highlights why MeZO can optimize large LMs, although a number of classical results [69, 47, 79, 3, 70] suggest that optimization should be catastrophically slow when training so many parameters. The inclusion of a simple prompt is crucial for MeZO to succeed (Appendix A). Past works [83, 67] have suggested that including such a prompt ensures that the fine-tuning objective is closely related to the pre-training one. As such, here, we make the assumption that the model has already been trained for many steps on the fine-tuning objective, which implies that the loss landscape exhibits favorable conditions (Assumption 1). Then, we derive a convergence rate independent of the number of parameters. We show that the loss decreases per step at a rate independent of the parameter dimension $d$ (Theorem 1), and that, under stronger conditions, the algorithm converges in time independent of $d$ (Lemma 3). Together, these results imply that MeZO is not catastrophically slower than SGD when fine-tuning.[6] For ease of illustration, we assume that $z$ is sampled from a sphere with radius $\sqrt{d}$, and in Appendix G.2, we derive the rate for a general Gaussian $z$, which was used in the experiments.

We follow classical analyses of SGD and replace the minibatch gradient estimate with SPSA (Definition 1). Consider the minibatch SGD update $\theta_{t+1} \leftarrow \theta_t - \eta \overline{\nabla} \mathcal{L}(\theta; \mathcal{B}_t)$ where $\mathcal{B}_t$ is a minibatch drawn uniformly from $\mathcal{D}^B$. Crucially, the SGD minibatch gradient estimate is unbiased.

**Definition 3** (Unbiased Gradient Estimate). *Any minibatch gradient estimate $g(\theta, \mathcal{B})$ is said to be unbiased if $\mathbb{E}[g(\theta, \mathcal{B})] = \nabla \mathcal{L}(\theta)$.*

## 4.1   Per-step analysis

The classical descent lemma uses a Taylor expansion to study how SGD reduces the loss at each optimization step. It highlights that when the gradient covariance is large, the maximum possible decrease in loss at each optimization step is small, thereby resulting in slower optimization.

**Lemma 1** (Descent Lemma). *Let $\mathcal{L}(\theta)$ be $\ell$-smooth.[7] For any unbiased gradient estimate $g(\theta, \mathcal{B})$,*

$$\mathbb{E}[\mathcal{L}(\theta_{t+1}) \mid \theta_t] - \mathcal{L}(\theta_t) \leq -\eta \left\| \nabla \mathcal{L}(\theta_t) \right\|^2 + \frac{1}{2} \eta^2 \ell \cdot \mathbb{E}[\| g(\theta, \mathcal{B}_t) \|^2]. \tag{2}$$

The descent lemma highlights the importance of the gradient norm, which we derive for MeZO below.

**Lemma 2.** *Let $\mathcal{B}$ be a random minibatch of size $B$. Then, the gradient norm of MeZO is*

$$\mathbb{E}_x \left[ \left\| \widehat{\nabla} \mathcal{L}(\theta; \mathcal{B}) \right\|^2 \right] = \frac{d + n - 1}{n} \mathbb{E} \left[ \left\| \nabla \mathcal{L}(\theta; \mathcal{B}) \right\|^2 \right].$$

*where $n$ is the number of $z$ sampled in $n$-SPSA (Definition 1) and $d$ is the number of parameters.*

Thus, in the usual case where $n \ll d$, MeZO has a much larger gradient norm than SGD.[8] The descent lemma also shows that to guarantee loss decrease, one needs to choose the learning rate as

$$\eta \leq \frac{2 \left\| \nabla \mathcal{L}(\theta_t) \right\|^2}{\ell \cdot \mathbb{E}[\| g(\theta, \mathcal{B}) \|^2]} \qquad \xRightarrow{\text{Lemma 2}} \qquad \eta_{\text{ZO}} = \frac{n}{d + n - 1} \eta_{\text{SGD}} \tag{3}$$

where $\eta_{\text{ZO}}$ and $\eta_{\text{SGD}}$ are the maximum permissible learning rates for MeZO and SGD respectively. Thus we see that without any further assumptions, MeZO can slow optimization by decreasing the largest permissible learning rate by a factor of $d$. Moreover, MeZO reduces the loss decrease that can be obtained at each step and, as a consequence, slows convergence by a factor of $d$ as well.

Surprisingly, our experiments show that MeZO can quickly optimize pre-trained models with billions of parameters, and reducing the number of tuned parameters via PEFT techniques does not substantially accelerate optimization (Appendix F.3). We attribute these phenomena to the Hessian of the loss exhibiting small local effective rank. It is prohibitively expensive to directly measure the effective rank of the Hessian of a large LM on a reasonably sized dataset. However, many previous works have shown that the Hessian of the loss for deep neural networks trained by SGD has remarkably low

---

[6]Section 3 uses the standard choice of Adam for FT; we provide SGD experiments in Appendix F.1.

[7]This is satisfied for the standard cross-entropy objective.

[8]All of our experiments use $n = 1$.

effective rank [74, 75, 36, 107, 105, 82]. In particular, the bulk of the spectrum concentrates around 0 with only a small number of outliers, and the number of these outliers is an upper bound on the effective rank. In addition, prior works [4, 56] have demonstrated that LM fine-tuning can occur in a very low dimensional subspace ($< 200$ parameters), which further supports the below assumption. We formalize the assumption on the effective rank below. In particular, we require an upper bound on the Hessian in a neighborhood around the current iterate to have effective rank at most $r$.

**Assumption 1** (Local $r$-effective rank). *Let $G(\boldsymbol{\theta}_t) = \max_{(\boldsymbol{x},\boldsymbol{y}) \in \mathcal{D}} \|\nabla \mathcal{L}(\boldsymbol{\theta}_t; \{(\boldsymbol{x},\boldsymbol{y})\})\|$. There exists a matrix $\boldsymbol{H}(\boldsymbol{\theta}_t) \preceq \ell \cdot \boldsymbol{I}_d$ such that:*

*1. For all $\boldsymbol{\theta}$ such that $\|\boldsymbol{\theta} - \boldsymbol{\theta}_t\| \leq \eta d G(\boldsymbol{\theta}_t)$, we have $\nabla^2 \mathcal{L}(\boldsymbol{\theta}) \preceq \boldsymbol{H}(\boldsymbol{\theta}_t)$.*

*2. The effective rank of $\boldsymbol{H}(\boldsymbol{\theta}_t)$, i.e $\mathrm{tr}(\boldsymbol{H}(\boldsymbol{\theta}_t))/\|\boldsymbol{H}(\boldsymbol{\theta}_t)\|_{op}$, is at most $r$.*

Under this assumption, we show that the convergence rate of ZO-SGD does not depend on the number of parameters. Instead, the slowdown factor only depends on the effective rank of the Hessian.

**Theorem 1** (Dimension-Free Rate). *Assume the loss exhibits local $r$-effective rank (Assumption 1). If $\boldsymbol{\theta}_{t+1} = \boldsymbol{\theta}_t - \eta_{ZO}\widehat{\nabla}\mathcal{L}(\boldsymbol{\theta}_t; \mathcal{B})$ is a single step of ZO-SGD using the $n$-SPSA estimate with a minibatch of size $B$, then there exists a $\gamma = \Theta(r/n)$ such that the expected loss decrease can be bounded as*

$$\mathbb{E}[\mathcal{L}(\boldsymbol{\theta}_{t+1}) \mid \boldsymbol{\theta}_t] - \mathcal{L}(\boldsymbol{\theta}_t) \leq -\eta_{ZO} \|\nabla\mathcal{L}(\boldsymbol{\theta}_t)\|^2 + \frac{1}{2}\eta_{ZO}^2 \ell \cdot \gamma \cdot \mathbb{E}[\|\nabla\mathcal{L}(\boldsymbol{\theta}; \mathcal{B})\|^2] \quad (4)$$

By applying Equation (3), we can directly compare to the SGD descent lemma.

**Corollary 1.** *Choosing the learning rate $\eta_{ZO} = \gamma^{-1} \cdot \eta_{SGD}$, ZO-SGD obtains a loss decrease of*

$$\mathbb{E}[\mathcal{L}(\boldsymbol{\theta}_{t+1}) \mid \boldsymbol{\theta}_t] - \mathcal{L}(\boldsymbol{\theta}_t) \leq \frac{1}{\gamma} \cdot \left[ -\eta_{SGD} \|\nabla\mathcal{L}(\boldsymbol{\theta}_t)\|^2 + \frac{1}{2}\eta_{SGD}^2 \ell \cdot \mathbb{E}[\|\nabla\mathcal{L}(\boldsymbol{\theta}; \mathcal{B})\|^2] \right]. \quad (5)$$

Here we see that comparing to SGD, the slowdown factor of ZO-SGD scales with the local effective rank $r$, which we argue is much smaller than the number of parameters $d$. The above analysis focuses on how much ZO-SGD and SGD decrease the loss at each step. Below, we show that under stronger assumptions about the loss landscape, we can obtain rates for how quickly the ZO-SGD algorithm converges to an optimal value.

## 4.2 Global convergence analysis

We show that the global convergence rate also slows by a factor proportional to the local effective rank under stronger assumptions about the loss landscape. We assume that the landscape obeys the classical PL inequality: the gradient norm grows quadratically with the suboptimality of the iterate.

**Definition 4** (PL Inequality). *Let $\mathcal{L}^* = \min_{\boldsymbol{\theta}} \mathcal{L}(\boldsymbol{\theta})$. The loss $\mathcal{L}$ is $\mu$-PL if, for all $\boldsymbol{\theta}$, $\frac{1}{2}\|\nabla\mathcal{L}(\boldsymbol{\theta})\|^2 \geq \mu(\mathcal{L}(\boldsymbol{\theta}) - \mathcal{L}^*)$.*

The PL inequality is not as strong as assuming that optimization exhibits kernel-like dynamics, but it ensures that the landscape is amenable to analysis [50]. In addition to the PL inequality, we assume the trace of the gradient covariance is bounded, so noise does not disrupt the trajectory too drastically.

**Definition 5** (Gradient Covariance). *The SGD gradient estimate on a minibatch of size $B$ has covariance $\boldsymbol{\Sigma}(\boldsymbol{\theta}) = B(\mathbb{E}\left[\nabla\mathcal{L}(\boldsymbol{\theta}; \mathcal{B})\nabla\mathcal{L}(\boldsymbol{\theta}; \mathcal{B})^\top\right] - \nabla\mathcal{L}(\boldsymbol{\theta})\nabla\mathcal{L}(\boldsymbol{\theta})^\top).*

As we show in Appendix G.1, this assumption holds for common loss functions such as square loss or binary cross entropy for several settings (e.g., kernel behavior [67]). With these two assumptions, we show that ZO-SGD has a slowdown proportional to the effective rank $r$, not the parameter dimension.

**Lemma 3** (Global Convergence of ZO-SGD). *Let $\mathcal{L}(\boldsymbol{\theta})$ be $\mu$-PL and let there exist $\alpha$ such that $\mathrm{tr}(\boldsymbol{\Sigma}(\boldsymbol{\theta})) \leq \alpha(\mathcal{L}(\boldsymbol{\theta}) - \mathcal{L}^*)$ for all $\boldsymbol{\theta}$. Then after*

$$t = \mathcal{O}\left( \left(\frac{r}{n} + 1\right) \cdot \underbrace{\left(\frac{\ell}{\mu} + \frac{\ell\alpha}{\mu^2 B}\right) \log \frac{\mathcal{L}(\boldsymbol{\theta}_0) - \mathcal{L}^*}{\epsilon}}_{\text{SGD rate (Lemma 4)}} \right)$$

*iterations of ZO-SGD we have $\mathbb{E}[\mathcal{L}(\boldsymbol{\theta}_t)] \leq \mathcal{L}^* + \epsilon$.*

# 5   Related work

**Zeroth-order optimization**   Many classical lower bounds have been derived for ZO-SGD in the strongly convex and convex settings [47, 3, 79, 32, 85, 69] as well as non-convex [101]. These bounds generally depended on the number of parameters $d$. More recently, [100, 6, 15] showed that if the gradient has low-dimensional structure, then the query complexity scales linearly with the intrinsic dimension and logarithmically with the number of parameters, though the estimation has at least $\Omega(sd \log d)$ memory cost. Additional tricks such as sampling schedules [11] and other variance reduction methods [48, 62] can be added to ZO-SGD. ZO has inspired distributed methods [93, 43] and black-box adversarial example generation [14, 63, 17, 64] in deep learning. Ye et al. [108], Balasubramanian and Ghadimi [7] estimate the Hessian to perform ZO optimization along important directions. There are also ZO methods that optimize without estimating the gradient [38, 68, 44].

**Memory-efficient backpropagation**   Several algorithms have been proposed to efficiently approximate backpropagation by sparsifying gradients [92, 102], approximating Jacobians [1, 19], and subsampling the computational graph [71, 2]. However, these methods may accrue large approximation errors for deep networks. Gradient checkpointing [18] reduces memory cost of backpropagation at the cost of recomputing some activations. FlashAttention [23] also reduces memory cost by recomputing attention matrices. Dettmers et al. [26, 27] explore quantization of large LMs' weights and optimizer states, which leads to memory reduction in both training and inference.

**Gradient-free adaptation of large language models**   BBT and BBTv2 [91, 90] use evolutionary algorithms to achieve gradient-free optimization; however, due to its sensitivity to high dimensionality, BBT is limited to only optimize a low-dimension projection of prefixes and they focus on RoBERTa-large size models and few-shot settings. Other works in "black-box tuning" of LMs focus on optimizing discrete prompts without updating the model, either via reinforcement learning [16, 25, 29], ensemble [45], or iterative search [78]. Concurrent work in [106] uses iterative forward passes to improve in-context learning performance.

# 6   Conclusion

We have shown that MeZO can effectively optimize large LMs across many tasks and scales. Further experiments suggest that MeZO can optimize non-differentiable objectives, which backpropagation usually cannot do. Our theory illustrates why MeZO is not catastrophically slow when tuning billions of parameters. As a limitation, MeZO takes many steps in order to achieve strong performance, though we show that the per-step speedup in MeZO can often make fine-tuning with MeZO run faster than a standard implementation of fine-tuning with backpropagation. We did not explore combining MeZO with other memory-efficient methods, such as FlashAttention [23] and quantization [26], though we hope to investigate this in the future.

We are excited to explore the applicability of MeZO to a number of areas, including but not limited to: pruning, distillation, saliency, interpretability, and dataset selection for fine-tuning. Non-differentiable objectives are a particularly exciting area, given recent advances in tuning large LMs to adapt to human feedback. Conducting theoretical analyses for how these efficient gradient estimates impact the performance of different applications is also of interest.

## Acknowledgements

We thank Xinyi Chen, Yin Tat Lee, Kaifeng Lyu, Tengyu Ma, Abhishek Panigrahi, Nikunj Saunshi, and Mengzhou Xia for their helpful feedback. SA and SM are funded by NSR, ONR, SRC, and Simons Foundation. JDL, AD, and EN acknowledge the support of the ARO under MURI Award W911NF-11-1-0304, the Sloan Research Fellowship, NSF CCF 2002272, NSF IIS 2107304, NSF CIF 2212262, ONR Young Investigator Award, and NSF CAREER Award 2144994. TG is supported by an IBM PhD Fellowship. This work is also partially funded by the National Science Foundation (IIS-2211779).

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

# A Algorithmic Ablations

We perform a number of ablations to select the best algorithm. As is standard in ZO literature, we consider the main computational cost to be the number of forward passes. In our case, the number of forward passes can be affected by the number of gradient steps taken, any usage of gradient accumulation, and using more noise samples to reduce the variance of the gradient estimate.

We observed that the performance of MeZO improves monotonically with the number of steps, and there does not appear to be any overfitting. Hence, when performing algorithmic ablations, we can focus on the efficiency of different algorithms without considering implicit bias. This is also reflected in our theoretical analysis. To ease the computational load, we fix the number of forward passes to $10,000$ and compare many different algorithms for RoBERTa-large on a smaller set of tasks that span sentiment analysis, entailment, and topic classification: SST-2, SNLI, and TREC. We emphasize that $10,000$ is a small budget and is only used as a setting to compare these ZO algorithms to each other. We find that using a linearly decreasing learning rate schedule during training, as was done for fine-tuning with backpropagation in [65], does not help or hurt MeZO. Similarly, using a learning rate warmup leads to identical results on these three tasks. For simplicity, we use a constant learning rate schedule with no warmup for all of the below experiments. We perform few-shot experiments with $k = 16$ and average the results across 5 seeds.

| Experiment | Hyperparameters | Values |
|---|---|---|
| MeZO | Batch size | $\{16, 64\} \times$ |
| | Learning rate | $\{1e-5, 1e-6, 1e-7\} \times$ |
| | $\epsilon$ | $\{1e-3, 1e-5\} \times$ |
| | Weight Decay | $\{0, 0.1\}$ |

Table 4: The hyperparameter grid used in our ablation experiments. For simplicity, we use a constant learning rate schedule.

## A.1 Prompting

We study if adding a prompt is crucial to the ability of MeZO to optimize the network. We use prompts from Gao et al. [35]. Malladi et al. [67] claimed the prompt makes the optimization trajectory well-behaved, though we note that the current paper considers RoBERTa-large and large autoregressive models while the previous work only studied RoBERTa-base. We note the similarity between kernel behavior and our theoretical setting in Section 4. MeZO succeeds on tasks that are reported to not exhibit kernel behavior in Malladi et al. [67], so we investigate whether or not the prompt is necessary.

| | SST-2 | SNLI | TREC |
|---|---|---|---|
| Prompt | 89.6 (1.2) | 65.1 (6.2) | 66.7 (6.2) |
| No Prompt | 51.9 (2.9) | 34.8 (2.1) | 19.5 (9.0) |

Table 5: Experiments using MeZO to fine-tune models with and without a prompt.

Both experiments followed the grid in Table 4, but we also expanded the grid to include a learning rate of $1e-4$ for the no prompt case. As a result of these experiments, we fix the setting to prompt-based fine-tuning for all of the below experiments.

## A.2 Sample schedules

One can sample $n_t$ noise vectors at the $t$th step and use $n_t$-SPSA to compute the gradient estimate. Similar ideas were proposed in Bollapragada et al. [11], Cai et al. [15]. We study the effect of linearly increasing and constant sampling schedules in the ablation setting. The intuition for the linearly increasing schedule is that the optimizer may need a higher fidelity gradient as it approaches the minimum. Increasing the number of $z$s can speed up optimization by reducing the gradient variance, but doing so also increases the number of forward passes required for each optimization step, so there is a trade-off to study. We note that increasing the number of $z$s should be accompanied by

a proportional scaling of the learning rate, analogous to the linear scaling rule proposed in [39] (theoretical justification can follow the SDE technique [58]). Table 6 shows no consistent benefit in one schedule over the other, and it demonstrates that increasing the $n$ in $n$-SPSA while fixing the number of forward passes allowed results in only marginal gains at best.

| $n$ | Schedule | SST-2 | SNLI | TREC |
|-----|----------|-------|------|------|
| $n = 1$ | Constant | 89.6 (1.2) | 65.1 (6.2) | **66.7 (6.2)** |
| $n = 4$ | Constant | 89.5 (1.1) | **68.6 (3.2)** | 62.3 (5.6) |
| $n = 4$ | Linear | 89.6 (1.4) | 65.3 (6.4) | 66.1 (5.5) |
| $n = 16$ | Constant | **90.4 (0.7)** | 67.0 (3.4) | 62.8 (6.3) |
| $n = 16$ | Linear | 88.9 (1.2) | 62.8 (5.9) | 64.2 (5.3) |

Table 6: Experiments using MeZO with different schedules for $n$. We scale the learning rate proportionally to the number of $z$'s sampled.

## B    MeZO Variants

There is a rich history of transferring ideas from first order optimization to enhance ZO algorithms. Below, we highlight several variants of MeZO that did not achieve as high performance as the algorithm presented in Algorithm 1.

### B.1    Memory-efficient n-SPSA

We highlight how MeZO can perform $n$-SPSA (Definition 1) efficiently for $n > 1$ in Algorithm 2. In particular, if sampling $n$ $z$ vectors and averaging the projected gradients, we require storing $2n$ additional scalars: the random seeds and the projected gradients. The same caveat about perturbing individual weights versus entire weight matrices still applies here (see Section 2).

### B.2    Augmenting MeZO with Gradient History

The $n$-SPSA algorithm merely provides a gradient estimate that can subsequently be used in place of the gradient in any gradient-based optimizer. Many popular optimizers, such as Adam and SGD with momentum, require storing some historical information about gradients (e.g., a moving average). This requirement causes such algorithms to require $2\times$ or $3\times$ the memory that is needed for SGD.

However, one advantage of MeZO is that the gradient history can be recomputed at each step without requiring much additional memory. In reference to Algorithm 1, note that the gradient only needs `projected_grad` and the random seed $s$ used to compute the perturbation $z$, so we need to only store 2 scalars per step to reproduce the gradient history (i.e., up to $2T$ scalars during training). This is a substantial reduction in added memory overhead that is usually needed for using Adam or momentum instead of vanilla SGD.

Table 18 illustrates that MeZO-Adam can sometimes improve the performance of MeZO, though each gradient step requires additional computation (but no additional forward passes). We leave it to future work to investigate when MeZO-Adam may be more useful than MeZO.

| Experiment | Hyperparameters | Values |
|------------|-----------------|--------|
| MeZO-Adam | Batch size | 64 |
| | Learning rate | $\{1e{-}6, 1e{-}5, 1e{-}4, 5e{-}4, 1e{-}3\}$ |
| | $\epsilon$ | $1e{-}3$ |
| | Weight Decay | 0 |

Table 7: The hyperparameter grid used for MeZO-Adam. For simplicity, we use a constant learning rate schedule.

**Algorithm 2:** MeZO with $n > 1$

---

**Require**: parameters $\boldsymbol{\theta} \in \mathbb{R}^d$, loss $\mathcal{L} : \mathbb{R}^d \to \mathbb{R}$, step budget $T$, perturbation scale $\epsilon$, batch size
   $B$, learning rate schedule $\{\eta_t\}$, $n$ for $n$-SPSA estimate (Definition 1)

---

**for** $t = 1, ..., T$ **do**
   seeds, projected_grads $\leftarrow$ []          $\triangleright$ Will each contain $n$ scalars
   **for** $j = 1, ..., n$ **do**
      Sample batch $\mathcal{B} \subset \mathcal{D}^B$ and random seed $s$
      $\boldsymbol{\theta} \leftarrow$ PerturbParameters($\boldsymbol{\theta}, \epsilon, s$)
      $\ell_+ \leftarrow \mathcal{L}(\boldsymbol{\theta}; \mathcal{B})$
      $\boldsymbol{\theta} \leftarrow$ PerturbParameters($\boldsymbol{\theta}, -2\epsilon, s$)
      $\ell_- \leftarrow \mathcal{L}(\boldsymbol{\theta}; \mathcal{B})$
      $\boldsymbol{\theta} \leftarrow$ PerturbParameters($\boldsymbol{\theta}, \epsilon, s$)         $\triangleright$ Reset parameters
      projected_grad $\leftarrow (\ell_+ - \ell_-)/(2\epsilon)$
      projected_grads[j] $\leftarrow$ projected_grad
      seeds[j] $\leftarrow s$
   **end**

   **for** $j = 1, ..., n$ **do**
      Reset random number generator with seed seeds[j]
      **for** $\theta_i \in \boldsymbol{\theta}$ **do**
         $z \sim \mathcal{N}(0, 1)$
         $\theta_i \leftarrow \theta_i - (\eta_t/n) *$ projected_grads[j] $* z$   $\triangleright$ Avg grad for $z_1, ..., z_n$
      **end**
   **end**
**end**

**Subroutine** PerturbParameters($\boldsymbol{\theta}$, $\epsilon$, $s$)
   Reset random number generator with seed $s$         $\triangleright$ For sampling $z$
   **for** $\theta_i \in \boldsymbol{\theta}$ **do**
      $z \sim \mathcal{N}(0, 1)$
      $\theta_i \leftarrow \theta_i + \epsilon z$         $\triangleright$ Modify parameters in place
   **end**
   **return** $\boldsymbol{\theta}$

---

### B.3   Modifying the Variance of MeZO

Our theory in Section 4 sketches the well-known fact that the variance of the stochastic gradient estimate can impact the rate of optimization. ZO methods can be combined with standard variance reduction techniques to possibly improve optimization speed. For example, Liu et al. [62] designed a variance reduced ZO algorithm, analogous to SVRG [49], to improve the speed of convergence. Below, we show that several variance reduction methods (e.g., using the gradient norm) can be implemented in a memory-efficient manner. However, when controlling for the total budget of forward passes (i.e., function queries), these methods are not as performant as MeZO. We nevertheless present them to demonstrate the ease with which MeZO can be adapted, and we suggest these methods may be useful for optimizing more complex objectives.

First, we define a general SPSA estimate that has the same expectation (i.e., the true gradient) but has a scaled variance.

**Definition 6** (Variance-Modified SPSA). *Given a matrix $D = \text{diag}(\boldsymbol{d})$, the variance modified SPSA computes*

$$\widetilde{\nabla}\mathcal{L}(\boldsymbol{\theta}; \mathcal{B}) = \frac{\mathcal{L}(\boldsymbol{\theta} + \epsilon(\boldsymbol{d}^{-1} \odot \boldsymbol{z}); \mathcal{B}) - \mathcal{L}(\boldsymbol{\theta} - \epsilon(\boldsymbol{d}^{-1} \odot \boldsymbol{z}); \mathcal{B})}{2\epsilon}(\boldsymbol{d} \odot \boldsymbol{z})$$

*where $\boldsymbol{d} \in \mathbb{R}^d$ has nonzero entries and $\boldsymbol{d}^{-1}$ denotes the coordinatewise reciprocal.*

The above SPSA variant is an unbiased estimator of the gradient, because $\mathbb{E}[\widetilde{\nabla}\mathcal{L}(\boldsymbol{\theta}; \mathcal{B})] = \mathbb{E}[D^{-1}\boldsymbol{z}\boldsymbol{z}^\top D \nabla \mathcal{L}(\boldsymbol{\theta}; \mathcal{B})] = \mathbb{E}[\nabla \mathcal{L}(\boldsymbol{\theta}; \mathcal{B})]$. We will draw inspiration from classical methods (i.e., "control variates") and choose $\boldsymbol{d}$ to be a block vector with gradient norms or parameter norms [99].

To select the parameter groups, we split the model by layer, keeping the embedding and the head separate (i.e., RoBERTa-large has $24 + 2 = 26$ parameter groups). It is straightforward to measure the parameter norms without consuming additional memory. We can measure the gradient norms without performing backpropagation, as shown below.

**Proposition 1** (ZO Estimate of Gradient Norm of $\ell$th Layer). *Define $z_\ell$ to have $z \sim \mathcal{N}(0, 1)$ in each coordinate corresponding to parameters in the $\ell$th layer and $0$ everywhere else. Then, we can estimate the norm of the gradient of the loss w.r.t. the $\ell$th layer $\nabla_{\theta_\ell}$ as*

$$\|\nabla_{\theta_\ell} \mathcal{L}(\theta; \mathcal{B})\|_2 \approx \left| \frac{\mathcal{L}(\theta + \epsilon z_\ell; \mathcal{B}) - \mathcal{L}(\theta - \epsilon z_\ell; \mathcal{B})}{2\epsilon} \right|$$

As is true for SPSA, increasing the number of $z_\ell$'s sampled for each value of $\ell$ and averaging the result reduces the variance of the estimate. The rationale for this estimate is that for any vector $v$, $\mathbb{E}_z[(\langle v, z \rangle)^2] = \|v\|_2^2$ for Gaussian $z$. It is clear that this estimate can be computed in a memory efficient way, although it requires $2L$ forward passes to compute gradient norms for $L$ parameter groups.

We show the experimental results for modifying the variance below. We follow the ablation setting and use a fixed budget of $10,000$ steps (Appendix A). Generally, using the gradient norm to reduce the variance substantially hurts performance (Table 8). If we "cheat" and allow one backpropagation through the network to estimate the gradient norm, then we see that reducing the variance using the gradient norm does not substantially hurt or help performance. Modifying the variance using the parameter norm, analogous to layerwise adaptive rate methods, does not substantially impact the performance of MeZO (Table 9).

Our observation is that decreasing the variance by setting $d$ as the gradient norm does not improve optimization. This empirical result agrees with the exposition in Section 4 that the straightforward variance analysis (which yields a dependence on the number of parameters $d$) is not the best lens to study the rate of optimization when fine-tuning with MeZO. Our effective rank view in Theorem 1 and Lemma 3 is likely a better characterization of fine-tuning dynamics. We leave it to future work to explore if these methods can be useful for other more complex objectives.

| Recompute $d$ | ZO estimate of $d$ | SST-2 | SNLI | TREC |
|---|---|---|---|---|
| Baseline MeZO (Algorithm 1) | | 89.6 (1.2) | 65.1 (6.2) | 66.7 (6.2) |
| ✗ | ✗ | 89.7 (0.8) | 65.2 (5.2) | 64.3 (6.4) |
| ✗ | ✓ | 87.0 (2.5) | 49.6 (9.2) | 32.6 (7.7) |
| ✓ | ✓ | 79.0 (10.3) | 48.9 (2.2) | 38.7 (7.5) |

Table 8: Experiments modifying the variance of MeZO using $d$ as the gradient norm (see Definition 6). We sometimes recompute $d$ at the start of each epoch or use Proposition 1 to estimate $d$ without requiring backpropagation.

| Recompute $d$ | SST-2 | SNLI | TREC |
|---|---|---|---|
| Baseline MeZO (Algorithm 1) | 89.6 (1.2) | 65.1 (6.2) | 66.7 (6.2) |
| ✗ | 89.2 (2.1) | 65.4 (4.2) | 64.8 (5.6) |
| ✓ | 88.2 (4.7) | 65.2 (4.0) | 64.7 (5.5) |

Table 9: Experiments modifying the variance of MeZO using $d$ as the parameter norm (see Definition 6). We sometimes recompute $d$ at the start of each epoch.

## B.4 Modifying the Expectation of MeZO

The above experiments show that modifying the variance of MeZO cannot consistently accelerate its convergence. However, a simple modification of Definition 6 allows us to change the expectation of MeZO as well. This can be used to efficiently estimate coordinate-wise normalized gradient-based optimizer updates (e.g., Adam).

**Definition 7** (Expectation-Modified SPSA). *Given a matrix $D = \mathrm{diag}(\boldsymbol{d})$, the variance modified SPSA computes*

$$\widetilde{\nabla}\mathcal{L}(\boldsymbol{\theta}; \mathcal{B}) = \frac{\mathcal{L}(\boldsymbol{\theta} + \epsilon(\boldsymbol{d}^{-1} \odot \boldsymbol{z}); \mathcal{B}) - \mathcal{L}(\boldsymbol{\theta} - \epsilon(\boldsymbol{d}^{-1} \odot \boldsymbol{z}); \mathcal{B})}{2\epsilon}\boldsymbol{z}$$

*where $\boldsymbol{d} \in \mathbb{R}^d$.*

Now, we see that $\widetilde{\nabla}\mathcal{L}(\boldsymbol{\theta}; \mathcal{B}) = \mathbb{E}[D^{-1}\boldsymbol{z}\boldsymbol{z}^\top \nabla\mathcal{L}(\boldsymbol{\theta}; \mathcal{B})]$ so the SPSA estimate is no longer an unbiased estimator for $\nabla\mathcal{L}(\boldsymbol{\theta})$. If we choose $\boldsymbol{d}$ to be the gradient norm, for example, then SPSA can estimate the normalized gradient. Concurrent work in Tang et al. [94] gives another ZO estimate of the normalized gradient while assuming access to only rankings of inputs (instead of the noisy function evaluations available in our setting). We find that estimating the normalized gradient does not perform as well as directly estimating the gradient (Table 10). Regardless, we present this algorithm as a way to highlight that any coordinate-wise operation to the gradient can be applied in a memory-efficient manner.

| Method | SST-2 | SNLI | TREC |
|---|---|---|---|
| Baseline MeZO (Algorithm 1) | 89.6 (1.2) | 65.1 (6.2) | 66.7 (6.2) |
| Estimate of normalized gradient (Definition 7) | 88.0 (1.2) | 60.0 (2.4) | 44.0 (14.0) |

Table 10: Experiments modifying the expectation of MeZO using $\boldsymbol{d}$ as the gradient norm (see Definition 7). We use the ZO estimate of the gradient norm (Proposition 1).

### B.5 One-point estimate

Here, we investigate the efficacy of one-point gradient estimators in place of the two-point SPSA method. Using a one-point estimator instead of SPSA can reduce the MeZO running time by half. Many one-point estimators have been proposed in the past [34, 87, 95]. For simplicity, we focus on one estimator [113] that has a form reminiscent of SPSA but requires only one forward pass to estimate the gradient at each step.

**Definition 8** (One-Point Gradient Estimate, Zhang et al. [113]). *For a loss function $\mathcal{L}$ evaluated on a batch $\mathcal{B}_t$ with parameters $\boldsymbol{\theta}_t$ at step $t$, we can draw random noise $\boldsymbol{z}_t \sim \mathcal{N}(0, I_d)$ and compute the gradient estimate using hyperparameter $\epsilon$ as written below.*

$$\widehat{\nabla}\mathcal{L}(\boldsymbol{\theta}_t; \mathcal{B}_t) = \frac{\mathcal{L}(\boldsymbol{\theta}_t + \epsilon\boldsymbol{z}_t; \mathcal{B}_t) - \mathcal{L}(\boldsymbol{\theta}_{t-1} + \epsilon\boldsymbol{z}_{t-1}; \mathcal{B}_{t-1})}{\epsilon}$$

Notably, this one-point gradient estimate uses the loss at the previous iterate instead of evaluating the loss again at the current iterate. As such, this estimator requires only one forward pass at each iterate to estimate the gradient. For well-behaved loss functions and slow-moving optimization, these two formulas are intuitively similar. However, Table 11 finds this estimator to be much less efficient than SPSA when fixing the number of forward passes.

| | Steps | SST-2 | SST-5 | SNLI | MNLI | RTE | TREC |
|---|---|---|---|---|---|---|---|
| | | —— sentiment —— | | —— natural language inference —— | | | — topic — |
| SPSA [88] | 20K | **92.8** (0.5) | **51.3** (0.9) | **82.9** (1.0) | **74.4** (0.8) | **76.7** (1.7) | **92.7** (0.6) |
| One-point estimate [113] | 20K | 90.0 (0.4) | 44.6 (2.0) | 70.1 (1.5) | 57.2 (0.9) | 64.1 (1.0) | 57.3 (5.7) |
| One-point estimate [113] | 40K | 91.8 (0.5) | 45.9 (1.7) | 74.4 (0.8) | 61.0 (1.0) | 68.7 (1.2) | 73.0 (3.1) |

Table 11: Comparison between SPSA and a one-point estimate Zhang et al. [113]. One-point estimate only does one forward pass per step, thus is twice as fast as two-point estimate per step. As such, the number of forward passes after 40K steps with the one-point estimate is the same as the number of forward passes with SPSA after 20K steps. The results show that two-point estimate is much more effective than one-point estimate.

## C   Memory Analysis

The compute-memory tradeoff of backpropagation is complex to analyze. Griewank and Walther [40] provides a rigorous theoretical treatment of the problem. We empirically measure the memory

consumption of different methods for commonly used large language models, but here we hope to provide a more rigorous comparison of different gradient estimation algorithms, independent of the software used to implement them. Below, we summarize some key points that may help readers to understand how the MeZO compute-memory tradeoff compares to backpropagation.

Given a network, the first step to perform backpropagation is to decompose the model into easily differentiable blocks. We note that this decomposition is not unique. For each block, one can choose to cache the resulting output during the forward pass (thereby consuming memory) or instead recompute the output when it is needed (thereby consuming compute). The below proposition, adapted from Rule 21 in Griewank and Walther [40], captures this tradeoff.

**Proposition 2** (Time-Memory Tradeoff for Backpropagation, Griewank and Walther [40]). *Consider a network containing $N$ bits. For any time-memory tradeoff hyperparameter $c = O(1)$, there exists a backpropagation algorithm that runs in time $O(cN)$ and consumes memory proportional to $O(N^{1/c})$.*

Grimm et al. [41] also gave sharp bounds for the memory-time product. Note that the popular gradient checkpointing [18] method allows one to tune $c$ with limited precision (i.e., one cannot always further split a differentiable block and observe savings). Experiments in Chen et al. [18] choose $c = 2$ to achieve $O(\sqrt{N})$ memory while consuming $O(2N)$ computation. In the extreme case, gradient checkpointing allows one to use $O(N \log N)$ computation and $O(\log N)$ memory.

MeZO always consumes $2N$ compute and $O(1)$ memory, so it is more compute-efficient at the same memory cost as gradient checkpointing. Our exposition in Section 2 discusses that we can perturb groups of parameters together to save time while consuming additional memory. However, we do not consider that variant here because it is somewhere in the middle of the compute-memory pareto curve, where we cannot reason about what backpropagation will do. In particular, MeZO can split groups differently than backpropagation can, since MeZO does not require that each parameter group is easily differentiable, so it is hard to compare the two algorithms along the entire pareto curve.

We also compare backpropagation for the $c = 1$ case (i.e., storing everything during the forward pass). When storing everything, backpropagation consumes $O(N)$ time and $O(N)$ memory. Hence, SPSA consumes slightly more time and substantially less memory than backpropagation at this end of the tradeoff.

Unlike gradient checkpointing, MeZO computes only an approximation of the gradient. This approximation is only useful for fine-tuning with a prompt, making it less broadly useful than gradient checkpointing. There are other methods that approximate the gradient with less memory consumption than gradient checkpointing (see the Related Work section), though it is unclear how the memory consumption of those algorithms compare to MeZO.

# D   Forward Auto-Differentiation

We discuss the merits of using forward auto-differentiation instead of two forward passes to construct a gradient estimate for fine-tuning. As $\epsilon \to 0$, the SPSA gradient estimate (Definition 1) can be written as $zz^\top \nabla \mathcal{L}(\theta; \mathcal{B})$. The term $z^\top \nabla \mathcal{L}(\theta; \mathcal{B})$ is a Jacobian-vector product (JVP), and it is well-known that this can be computed in parallel with a single forward pass while consuming additional memory equivalent to that of the largest activation in the model. To fully compute the gradient estimate, one must store $z$ on the GPU while performing inference, so we observe that this algorithm requires more memory than MeZO.

We note that implementation of the forward auto-differentiation algorithm is not well-supported in PyTorch at the time of writing. The autograd JVP function computes the JVP in a memory-inefficient way, as noted in the documentation, and the other available methods to compute a JVP are not sophisticated enough to easily scale to a complex LLM. Computing the JVP is straightforward when using JAX, so we profile the memory consumption of inference and the JVP for RoBERTa-large when using JAX. We use batch size 16 with the MultiRC task. Note that JAX may automatically use rematerialization to avoid out of memory errors so we focus on settings in which the memory utilization remains below 50%. The resulting memory usage during inference, backpropagation, and forward auto-differentiation are reported in Table 12.

We see that forward auto-differentiation is substantially more memory efficient than backpropagation but less memory efficient than inference. Furthermore, forward auto-differentiation selects $\epsilon = 0$, which removes potentially beneficial third-and-higher order Taylor expansion terms from the estimate.

| Task | Inference (and **MeZO**) | Backpropagation | Forward Auto-Differentiation |
|---|---|---|---|
| Excess Memory (MB) | 327.50 | 24156.23 | 830.66 |

Table 12: Memory consumption of RoBERTa-large when using batch size 16 with the MultiRC task. The reported memory does not include the cost of storing the model on the GPU, which is required for all three cases.

## E   Experiment setup

### E.1   Datasets

For RoBERTa-large, we consider classification datasets: SST-2 [86], SST-5 [86], TREC [97], MNLI [103], SNLI [12], and RTE [22, 8, 37, 10]. We follow Malladi et al. [67] in limiting the test set to $1,000$ examples for fast iteration. For training and validation, we have two settings: $k = 16$ and $k = 512$, which mean that we have 16 or 512 examples per class for both training and validation.

For OPT experiments, we consider the SuperGLUE dataset collection [98], including: BoolQ [21], CB [24], COPA [81], MultiRC [51], ReCoRD [111], RTE [22, 8, 37, 10], WiC [77], and WSC [55]. We also include SST-2 [86] and two question answering (QA) datasets, SQuAD [80] and DROP [31]. We randomly sample 1,000 examples for training, 500 examples for validation, and 1,000 examples for testing.

### E.2   Prompts

Table 13 shows the set of downstream tasks and prompts with which we fine-tune RoBERTa-large, which are adapted from [35].

| Dataset | $C$ | Type | Prompt | Label words |
|---|---|---|---|---|
| SST-2 | 2 | sentiment cls. | `<`$S_1$`>` It was `[MASK]` . | {great, terrible} |
| SST-5 | 5 | sentiment cls. | `<`$S_1$`>` It was `[MASK]` . | {great, good, okay, bad, terrible} |
| TREC | 6 | topic cls. | `[MASK]` : `<`$S_1$`>` | {Description, Expression, Entity, Human, Location, Number} |
| MNLI | 3 | NLI | `<`$S_1$`>` ? `[MASK]` , `<`$S_2$`>` | {Yes, Maybe, No} |
| SNLI | 3 | NLI | `<`$S_1$`>` ? `[MASK]` , `<`$S_2$`>` | {Yes, Maybe, No} |
| RTE | 2 | NLI | `<`$S_1$`>` ? `[MASK]` , `<`$S_2$`>` | {Yes, No} |

Table 13: The prompts of the datasets we used in our RoBERTa-large experiments (Table 18 and Figure 2). The prompts are adapted from [35] and include a template and a set of label words that can fill in the `[MASK]` token. `<`$S_1$`>` and `<`$S_2$`>` refer to the first and the second (if any) input sentence.

Table 14 demonstrates the prompts we use for OPT. Note that in OPT experiments we have three types of tasks: classification, multiple-choice, and question answering. Prompts are adopted from GPT-3 [13] and PromptSource with minor changes [5].

### E.3   Hyperparameters

We use the hyperparameters in Table 15 for MeZO experiments on RoBERTa-large (Table 18 and Figure 2). Experiments in Appendix A informed the grid; in particular, the choice of $\epsilon$ seemed to not significantly impact performance, and using a larger batch size consistently yielded faster optimization. We use the hyperparameters in Table 16 for MeZO experiments on OPT.

Regarding learning rate scheduling and early stopping, we use linear learning scheduling for all fine-tuning with backpropagation experiments and constant learning rate for all MeZO experiments. For RoBERTa experiments, we evaluate the model on validation sets every 1/10 of total training steps and save the best validation checkpoint. All FT experiments use 1K steps and MeZO experiments use

| Dataset | Type | Prompt |
|---------|------|--------|
| SST-2 | cls. | `<text>` It was terrible/great |
| RTE | cls. | `<premise>`
Does this mean that "`<hypothesis>`" is true? Yes or No?
Yes/No |
| CB | cls. | Suppose `<premise>` Can we infer that "`<hypothesis>`"? Yes, No, or Maybe?
Yes/No/Maybe |
| BoolQ | cls. | `<passage>` `<question>`?
Yes/No |
| WSC | cls. | `<text>`
In the previous sentence, does the pronoun "`<span2>`" refer to `<span1>`? Yes or No?
Yes/No |
| WIC | cls. | Does the word "`<word>`" have the same meaning in these two sentences? Yes, No?
`<sent1>`
`<sent2>`
Yes/No |
| MultiRC | cls. | `<paragraph>`
Question: `<question>`
I found this answer "`<answer>`". Is that correct? Yes or No?
Yes/No |
| COPA | mch. | `<premise>` so/because `<candidate>` |
| ReCoRD | mch. | `<passage>`
`<query>`.replace("@placeholder", `<candidate>`) |
| SQuAD | QA | Title: `<title>`
Context: `<context>`
Question: `<question>`
Answer: |
| DROP | QA | Passage: `<context>`
Question: `<question>`
Answer: |

Table 14: The prompts of the datasets we used in our OPT experiments. There are three types of tasks: classification (cls.), multiple-choice (mch.), and question answering (QA). Prompts are adopted from GPT-3 [13] and PromptSource [5] with minor changes. `<text>` represents input from the dataset and Yes represents label words. For inference on multiple choice tasks, we put in different candidates in the prompt and calculate the average log-likelihood for each candidate, and choose the candidate with the highest score. For inference on QA tasks, we use greedy decoding to generate the answer.

100K steps. For OPT experiments, we evaluate the model on validation sets every 1/5 of the total training steps and save the best validation checkpoint. All FT experiments train for 5 epochs and all MeZO experiments use 20K steps. Note that FT experiments mostly converge within 5 epochs but we observe that MeZO performance can still improve with more training steps.

### E.4 Modeling and implementation

For RoBERTa experiments, we follow [35] for the prompt-based fine-tuning paradigm for masked language models. Please refer to the original paper for more details.

In OPT experiments, for classification tasks, we train the model similarly to [35], i.e., we take the logits corresponding to the label words and apply cross entropy loss on them; for multiple choice tasks and generation tasks (QA), we only keep the correct candidate and use teacher forcing to train on the correct examples. We only keep the loss on tokens in the candidate part and exclude the prompt part.

For OPT inference on classification and multiple-choice tasks, we use the model to get the average log-likelihood (by tokens) of all the candidates/label words, and predict the one with the highest average log-likelihood. For generation tasks, we use greedy decoding to generate the answer.

For in-context learning, we use 32 examples in the context. We also try filling in as many examples as possible in the context but this does not improve performance and sometimes leads to unstable results. Thus we keep the 32-example results.

| Experiment | Hyperparameters | Values |
|---|---|---|
| MeZO | Batch size | 64 |
| | Learning rate | $\{1e{-}7, 1e{-}6, 1e{-}5\}$ |
| | $\epsilon$ | $1e{-}3$ |
| | Weight Decay | 0 |
| MeZO (prefix) | Batch size | 64 |
| | Learning rate | $\{1e{-}2, 5e{-}3, 1e{-}3\}$ |
| | $\epsilon$ | $1e{-}1$ |
| | Weight Decay | 0 |
| | # prefix tokens | 5 |
| MeZO (LoRA) | Batch size | 64 |
| | Learning rate | $\{1e{-}5, 5e{-}5, 1e{-}4\}$ |
| | $\epsilon$ | $1e{-}3$ |
| | Weight Decay | 0.1 |
| | $(r, \alpha)$ | $(8, 16)$ |
| FT with Adam | Batch size ($k = 16$) | $\{2, 4, 8\}$ |
| | Batch size ($k = 512$) | $\{8, 16, 32\}$ |
| | Learning Rates | $\{1e{-}5, 3e{-}5, 5e{-}5\}$ |
| | Weight Decay | 0 |
| FT with SGD | Batch size ($k = 16$) | $\{2, 4, 8\}$ |
| | Batch size ($k = 512$) | $\{8, 16, 32\}$ |
| | Learning Rates | $\{1e{-}4, 5e{-}4, 1e{-}3, 5e{-}3, 1e{-}2\}$ |
| | Weight Decay | 0 |
| FT (prefix) | Batch size | $\{8, 16, 32\}$ |
| | Learning Rates | $\{1e{-}2, 3e{-}2, 5e{-}2\}$ |
| | Weight Decay | 0 |
| | # prefix tokens | 5 |
| FT (LoRA) | Batch size | $\{4, 8, 16\}$ |
| | Learning Rates | $\{1e{-}4, 3e{-}4, 5e{-}4\}$ |
| | $(r, \alpha)$ | $(8, 16)$ |

Table 15: The hyperparameter grids used for RoBERTa-large experiments. MeZO uses a constant learning rate schedule, and FT uses linear scheduling. All FT experiments use 1K steps and MeZO experiments use 100K steps. We check validation performance every 1/10 total training steps.

For linear probing of classification tasks, we take the output feature and use `scipy` package to train a linear classifier. For multiple-choice tasks and generation tasks, we found that this leads to poor results since the output space is the whole vocabulary; instead, we do head-tuning, where the whole model is fixed except for the LM projection head. We use a batch size of 8 and a learning rate of $\{1e{-}4\ 5e{-}4\}$, and train the head for 5 epochs.

For experiments on 30B and 66B OPT models, we largely follow the OPT hyperparameters except that we do not evaluate the intermediate validation performance and directly use the last checkpoint for evaluation, due to the high storage cost of intermediate checkpoints of large models.

### E.5 Parameter-efficient fine-tuning

Fine-tuning and storing a copy of the large language model for each downstream task is expensive. Parameter-efficient fine-tuning (PEFT) techniques alleviate this problem: instead of tuning all model parameters, PEFT only tunes a small number of additional parameters (usually less than 1%) and can often achieve comparable or better performance [57, 54, 30]. The ZO optimizer is compatible with PEFT methods, since ZO can operate on any subset of the model parameters. We are interested in the following two common PEFT methods, designed for transformers [96].

**LoRA** [46] adds a tunable low-rank delta to a linear layer during fine-tuning. Suppose a linear layer performed $\boldsymbol{W}\boldsymbol{x} + \boldsymbol{b}$ during pre-training with $\boldsymbol{W} \in \mathbb{R}^{m \times n}$. When fine-tuning, LoRA introduces two smaller matrices $\boldsymbol{A} \in \mathbb{R}^{m \times r}$ and $\boldsymbol{B} \in \mathbb{R}^{r \times n}$ such that $r \ll \min(m, n)$. The linear layer is then

| Experiment | Hyperparameters | Values |
|---|---|---|
| MeZO | Batch size | 16 |
| | Learning rate | $\{1e{-}6, 1e{-}7\}$ |
| | $\epsilon$ | $1e{-}3$ |
| MeZO (prefix) | Batch size | 16 |
| | Learning rate | $\{1e{-}2, 1e{-}3\}$ |
| | $\epsilon$ | $1e{-}1$ |
| | # prefix tokens | 5 |
| MeZO (LoRA) | Batch size | 16 |
| | Learning rate | $\{1e{-}4, 5e{-}5\}$ |
| | $\epsilon$ | $1e{-}2$ |
| | $(r, \alpha)$ | $(8, 16)$ |
| FT with Adam | Batch size | 8 |
| | Learning Rates | $\{1e{-}5, 5e{-}5, 8e{-}5\}$ |

Table 16: The hyperparameter grids used for OPT experiments. All weight decay is set to $0$. FT uses 5 epochs and linear scheduled learning rates and MeZO uses 20K steps and constant learning rates. We check validation performance and save the best checkpoint every 1/5 total training steps.

computed as

$$\left(\boldsymbol{W} + \frac{\alpha}{r}\boldsymbol{AB}\right)\boldsymbol{x} + \boldsymbol{b} \tag{6}$$

where $r$ and $\alpha$ are hyperparameters. $\boldsymbol{A}$ and $\boldsymbol{B}$ are trained on the downstream task while $\boldsymbol{W}$ is frozen at its pre-trained value. In transformers, this modification to the linear layer is applied to the query and value operations of each attention layer. Empirically, $r$ can be very small, so the number of trainable parameters during fine-tuning is small. We choose $r = 8$ and $\alpha = 16$.

**Prefix-tuning** [57] adds a prefix of $m$ tunable representations at each layer and freezes the rest of the model. The representations are added as new keys and values and treated as additional context during the attention operation. We initialize these tunable representations by randomly sampling tokens from the vocabulary and passing them through the LLM to get their keys and values at different attention layers. We found this crucial to make prefix tuning stable with MeZO, and this trick additionally boosts the performance of prefix tuning with backpropagation, as shown in Table 17. We also tried the reparameterization trick in [57], which does not help MeZO training. In our experiments, we find $m = 5$ to be sufficient to achieve good performance on most tasks.

We also show that MeZO is compatible with parameter-efficient fine-tuning methods, such as prefix tuning and LoRA. Surprisingly, the performance of MeZO does not improve substantially when tuning much fewer parameters, as one might expect from classical analyses (see Section 4). Accordingly, our theoretical analysis in Section 4 suggests that the convergence rate of ZO-SGD does not depend on the parameter dimension during fine-tuning.

| Task | SST-2 | SST-5 | SNLI | MNLI | RTE | TREC |
|---|---|---|---|---|---|---|
| Type | —— sentiment —— | | —— natural language inference —— | | | — topic — |
| FT (prefix, random init) | 90.7 (1.7) | 47.2 (2.0) | 70.7 (2.8) | 62.6 (3.3) | 63.5 (4.4) | 83.4 (4.7) |
| FT (prefix, real act init) | 91.9 (1.0) | 47.7 (1.1) | 77.2 (1.3) | 66.5 (2.5) | 66.6 (2.0) | 85.7 (1.3) |

Table 17: Prefix-tuning ablations. We compare randomly-initialized prefixes and real word activation prefixes. Using real word activations significantly outperforms random initialization.

### E.6 Training with non-differentiable objectives

The experiments maximizing the accuracy of a RoBERTa-large model were all conducted using the same grid as MeZO in Table 15.

For OPT experiments on SQuAD with F1 as objective, we use a batch size of 16. For MeZO, we use a learning rate of $\{1e{-}6, 5e{-}6, 1e{-}5\}$ and $\epsilon = 1e{-}3$. For MeZO (prefix), we use a learning rate of $\{1e{-}1, 5e{-}2, 1e{-}2\}$ and $\epsilon = 1e{-}1$.

### E.7 Memory profiling

In memory profiling, we use standard implementation with Huggingface's `transformers` [104] package. We did not turn on any advance memory-saving options, e.g., gradient checkpointing. We set the per-device batch size as 1 to test the minimum hardware requirement to run the model with specific optimization algorithms. For multi-GPU backpropagation, we use fully sharded data parallel (FSDP) [33] provided by PyTorch [76]. For multi-GPU MeZO, we use `transformers` multi-GPU inference of large models. We use Nvidia's `nvidia-smi` command to monitor the GPU memory usage. We call a run "successful" if there is no out of memory error from GPUs for at least 100 steps. We also profile fine-tuning with LoRA, but find its memory usage similar to that of fine-tuning with prefix-tuning. Hence here we only show the analysis with prefix-tuning.

## F  More experiment results

### F.1  RoBERTa-large experiments

Table 18 contains the detailed numbers corresponding to Figure 2 and also reports the performance of MeZO-Adam.

| Task | SST-2 | SST-5 | SNLI | MNLI | RTE | TREC |
|---|---|---|---|---|---|---|
| Type | —— sentiment —— | | —— natural language inference —— | | | — topic — |
| Zero-shot | 79.0 | 35.5 | 50.2 | 48.8 | 51.4 | 32.0 |
| *Gradient-free methods: $k = 16$* | | | | | | |
| LP | 76.0 (2.8) | 40.3 (1.9) | 66.0 (2.7) | 56.5 (2.5) | 59.4 (5.3) | 51.3 (5.5) |
| MeZO | 90.5 (1.2) | 45.5 (2.0) | 68.5 (3.9) | 58.7 (2.5) | 64.0 (3.3) | 76.9 (2.7) |
| MeZO (LoRA) | 91.4 (0.9) | 43.0 (1.6) | 69.7 (6.0) | 64.0 (2.5) | 64.9 (3.6) | 73.1 (6.5) |
| MeZO (prefix) | 90.8 (1.7) | 45.8 (2.0) | 71.6 (2.5) | 63.4 (1.8) | 65.4 (3.9) | 80.3 (3.6) |
| MeZO-Adam | 90.4 (1.4) | 45.4 (1.5) | 74.1 (2.7) | 64.3 (0.8)† | 59.2 (11.1)† | 78.3 (1.4) |
| *Gradient-based methods: $k = 16$* | | | | | | |
| FT | 91.9 (1.8) | 47.5 (1.9) | 77.5 (2.6) | 70.0 (2.3) | 66.4 (7.2) | 85.0 (2.5) |
| FT (LoRA) | 91.4 (1.7) | 46.7 (1.1) | 74.9 (4.3) | 67.7 (1.4) | 66.1 (3.5) | 82.7 (4.1) |
| FT (prefix) | 91.9 (1.0) | 47.7 (1.1) | 77.2 (1.3) | 66.5 (2.5) | 66.6 (2.0) | 85.7 (1.3) |
| *Gradient-free methods: $k = 512$* | | | | | | |
| LP | 91.3 (0.5) | 51.7 (0.5) | 80.9 (1.0) | 71.5 (1.1) | 73.1 (1.5) | 89.4 (0.5) |
| MeZO | 93.3 (0.7) | 53.2 (1.4) | 83.0 (1.0) | 78.3 (0.5) | 78.6 (2.0) | 94.3 (1.3) |
| MeZO (LoRA) | 93.4 (0.4) | 52.4 (0.8) | 84.0 (0.8) | 77.9 (0.6) | 77.6 (1.3) | 95.0 (0.7) |
| MeZO (prefix) | 93.3 (0.1) | 53.6 (0.5) | 84.8 (1.1) | 79.8 (1.2) | 77.2 (0.8) | 94.4 (0.7) |
| MeZO-Adam | 93.3 (0.6) | 53.9 (0.8) | 85.3 (0.8) | 79.6 (0.4) | 79.2 (1.2) | 95.1 (0.3) |
| *Gradient-based methods: $k = 512$* | | | | | | |
| FT | 93.9 (0.7) | 55.9 (0.9) | 88.7 (0.8) | 84.4 (0.8) | 82.7 (1.4) | 97.3 (0.2) |
| FT (LoRA) | 94.2 (0.2) | 55.3 (0.7) | 88.3 (0.5) | 83.9 (0.6) | 83.2 (1.3) | 97.0 (0.3) |
| FT (prefix) | 93.7 (0.3) | 54.6 (0.7) | 88.3 (0.7) | 83.3 (0.5) | 82.5 (0.8) | 97.4 (0.2) |

Table 18: Experiments on RoBERTa-large (350M parameters). LP: Linear probing; ZO, ZO (LoRA), and ZO (prefix): our memory-efficient ZO-SGD (Section 2.1) with full-parameter tuning, LoRA, and prefix-tuning respectively; FT: fine-tuning with Adam. All reported numbers are averaged accuracy (standard deviation). All experiments use prompts (Appendix E.2). ZO outperforms zero-shot and LP by a large margin and approaches FT performance with much less memory cost.

**LP-MeZO**  We also compare MeZO to performing linear probing and then subsequently performing fine-tuning via MeZO, following the analogous suggestion for fine-tuning in Kumar et al. [53]. We use the MeZO grid described in Table 15. Note that the linear probing checkpoints used here have early stopping, unlike the ones reported in Table 18. We heuristically implement early stopping by limiting the number of iterations (from 5000 to 1000) and increasing the convergence tolerance (from $1e-4$ to 0.01) in the `scipy` solver. Experiments on a few settings show that LP-MeZO can sometimes improve performance without increasing the memory consumption (see Table 19). However, sometimes, linear probing first can severely hurt performance.

| Task | SST-2 | SST-5 | SNLI | TREC |
|------|-------|-------|------|------|
| Zero-shot | 79.0 | 35.5 | 50.2 | 32.0 |
| FT | 91.9 (1.8) | 47.5 (1.9) | 77.5 (2.6) | 85.0 (2.5) |
| MeZO | 90.5 (1.2) | **45.5 (2.0)** | 68.5 (3.9) | **76.9 (2.7)** |
| LP-MeZO | **91.4 (1.4)** | 41.9 (3.3) | **70.7 (3.4)** | 54.0 (4.5) |

Table 19: Performing linear probing before fine-tuning with MeZO, as suggested previously [53], can sometimes improve performance without increasing the memory overhead. We use $k = 16$ for these experiments.

## F.2 OPT experiments

Table 20 present the full results of OPT-30B and OPT-66B, with detailed MeZO numbers.

| Task | SST-2 | RTE | BoolQ | WSC | WIC | SQuAD |
|------|-------|-----|-------|-----|-----|-------|
| 30B zero-shot | 56.7 | 52.0 | 39.1 | 38.5 | 50.2 | 46.5 |
| 30B ICL | 81.9 | 66.8 | 66.2 | 56.7 | 51.3 | 78.0 |
| 30B MeZO | 90.6 | 66.4 | 67.2 | 63.5 | 56.3 | 85.2 |
| 30B MeZO (prefix) | 87.5 | 72.6 | 73.5 | 55.8 | 59.1 | 83.9 |
| 66B zero-shot | 57.5 | **67.2** | 66.8 | 43.3 | 50.6 | 48.1 |
| 66B ICL | 89.3 | 65.3 | 62.8 | 52.9 | 54.9 | 81.3 |
| 66B MeZO | 91.2 | 65.7 | 72.7 | 63.5 | 58.9 | * |
| 66B MeZO (prefix) | 93.6 | 66.4 | 73.7 | 57.7 | 58.6 | 85.0 |

Table 20: Experiments on OPT-30B and OPT-66B (with 1,000 examples). *: MeZO requires further tuning to successfully optimize.

## F.3 Convergence of MeZO with full-parameter and PEFT

We demonstrate the convergence rate of MeZO, MeZO (LoRA) and MeZO (prefix) on SST-2 and SNLI for the first 5,000 steps in Figures 5. We see that despite the different number of parameters they optimize, MeZO demonstrates similar training speed on full parameter and PEFT. This agrees with our theory in Section 4, which shows that MeZO's optimization speed is independent of the number of parameters.

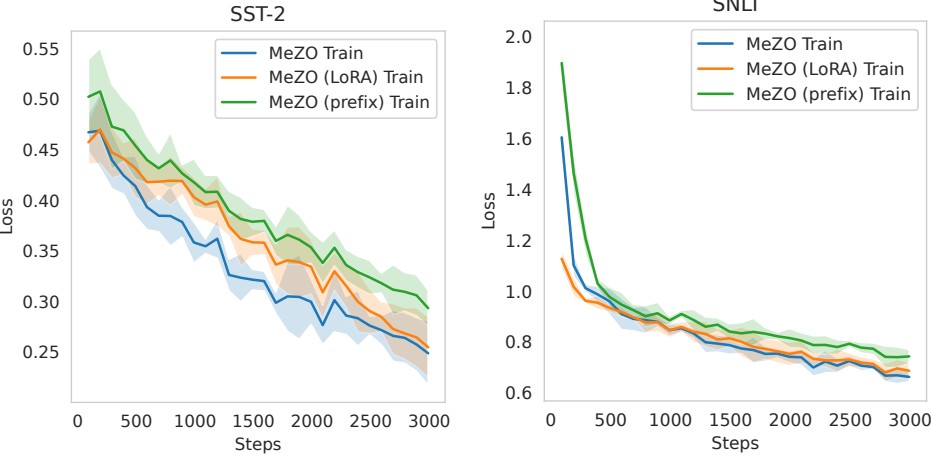

Figure 5: MeZO does not optimize significantly faster when tuning fewer parameters, agreeing with our theory in Section 4.

## F.4 ZO vs BBTv2

We compare ZO with BBTv2 [90] on mutually assessed tasks in Table 21. ZO significantly outperform BBTv2. Furthermore, BBTv2 is limited to optimize in low-dimensional space and requires prefix-tuning and a down-projection to reduce the number of optimized parameters. BBTv2 also employs an iterative scheme which only optimizes one layer at a time. In contrast, ZO works with both full-parameter tuning and PEFT, as shown in our experiments (Section 3) and theory (Section 4).

| Task | SST-2 | SNLI | RTE |
|---|---|---|---|
| Task type | —— sentiment —— | – natural language inference – | |
| Zero-shot | 79.0 | 50.2 | 51.4 |
| BBTv2 | 90.3 (1.7) | 57.3 (2.3) | 56.7 (3.3) |
| MeZO | 90.5 (1.2) | 68.5 (3.9) | 64.0 (3.3) |
| MeZO (LoRA) | 91.4 (0.9) | 69.7 (6.0) | 64.9 (3.6) |
| MeZO (prefix) | 90.8 (1.7) | 71.6 (2.5) | 65.4 (3.9) |

Table 21: ZO vs BBTv2 with RoBERTa-large. BBTv2 performance is from Sun et al. [90].

## F.5 Memory profiling

We show the detailed numbers of memory profiling results Table 22, which also corresponds to Figure 3. For how we profile the memory usage, please refer to Appendix E.7.

| Method | Zero-shot / MeZO | ICL | Prefix FT | Full-parameter FT |
|---|---|---|---|---|
| 1.3B | 1xA100 (4GB) | 1xA100 (6GB) | 1xA100 (19GB) | 1xA100 (27GB) |
| 2.7B | 1xA100 (7GB) | 1xA100 (8GB) | 1xA100 (29GB) | 1xA100 (55GB) |
| 6.7B | 1xA100 (14GB) | 1xA100 (16GB) | 1xA100 (46GB) | 2xA100 (156GB) |
| 13B | 1xA100 (26GB) | 1xA100 (29GB) | 2xA100 (158GB) | 4xA100 (316GB) |
| 30B | 1xA100 (58GB) | 1xA100 (62GB) | 4xA100 (315GB) | 8xA100 (633GB) |
| 66B | 2xA100 (128GB) | 2xA100 (134GB) | 8xA100 | 16xA100 |

Table 22: Memory usage on the MultiRC (avg #tokens=400) dataset.

## F.6 Wallclock time efficiency

In this section, we measure the wallclock time efficiency of MeZO compared to full-parameter FT, with respect to different model sizes. We conduct our experiments with 80GB A100s connected by NVLink and InfiniteBand, which are state-of-the-art solutions for distributed training. As shown in Table 23, on the MultiRC datasets, training with MeZO brings $7.74\times$ speedup per step compared to full-parameter FT on a 30B model. This is due to (1) MeZO does not require costly backpropagation and (2) MeZO requires fewer GPUs and reduces the multi-GPU communication overhead. We can see that MeZO has a bigger advantage when training larger models—the multi-GPU overhead for fine-tuning is larger.

Note that even though MeZO has better per-step wallclock efficiency, it requires significantly more steps than standard FT. Taking our OPT-30B experiments as an example: MeZO takes $32\times$ more steps than standard FT, while FT takes $8\times$ more GPUs and $7.74\times$ more time per step. Overall, MeZO requires only half as many GPU-hours as FT for a 30B model.

|            | 1.3B            | 2.7B            | 13B             | 30B             | 66B             |
|------------|-----------------|-----------------|-----------------|-----------------|-----------------|
| MeZO (bsz=16) | 0.815s (1)   | 1.400s (1)      | 2.702s (1)      | 5.896s (1)      | 12.438s (4)     |
| MeZO (bsz=8)  | 0.450s (1)   | 0.788s (1)      | 1.927s (1)      | 4.267s (1)      | 7.580s (2)      |
| FT (bsz=8)    | 0.784s (1)   | 1.326s (1)      | 13.638s (4)     | 45.608s (8)     | 84.098s (20)    |
|               | bspd=2, ga=4 | bspd=2, ga=4    | bspd=1, ga=2    | bspd=1, ga=1    | bspd=1, ga=1    |

Table 23: Wallclock time per step of different training methods. Numbers in brackets are numbers of GPUs required. It is measured on 80GB A100s with NVLink and InfiniteBand connections. The wallclock time is averaged over 100 steps. It is measured on the MultiRC task with the OPT family. We use a batch size ("bsz") of 8 for FT and 16 for MeZO (consistant with our main experiment setting). For comparison we also add MeZO with a batch size of 8. For FT (FSDP), we show the following additional information. "bspd": batch size per device. "ga": gradient accumulation steps. The effective batch size is bspd×ga× #GPUs. Note that for FT 66B, the effective batch size 20.

# G Proofs

*Proof of Lemma 2.* We first note that in the $\epsilon \to 0$ limit, we have

$$\widehat{\nabla}\mathcal{L}(\boldsymbol{\theta}; \mathcal{B}) = \frac{1}{Bn} \sum_{(\boldsymbol{x}, \boldsymbol{y}) \in \mathcal{B}} \sum_{i \in [n]} \boldsymbol{z}_i \boldsymbol{z}_i^\top \nabla \mathcal{L}(\boldsymbol{\theta}; \{(\boldsymbol{x}, \boldsymbol{y})\}).$$

Taking expectation over the batch $\mathcal{B}$ and the $\boldsymbol{z}_i$, we have $\mathbb{E}[\widehat{\nabla}\mathcal{L}(\boldsymbol{\theta}; \mathcal{B})] = \nabla \mathcal{L}(\boldsymbol{\theta})$, so $\widehat{\nabla}\mathcal{L}(\boldsymbol{\theta}; \mathcal{B})$ is an unbiased estimator of the gradient.

Computing the second moment, we get

$$\mathbb{E}\left[\widehat{\nabla}\mathcal{L}(\boldsymbol{\theta}; \mathcal{B})\widehat{\nabla}\mathcal{L}(\boldsymbol{\theta}; \mathcal{B})^\top\right]$$
$$= \frac{1}{B^2 n^2} \sum_{(\boldsymbol{x}_1, \boldsymbol{y}_1), (\boldsymbol{x}_2, \boldsymbol{y}_2) \in \mathcal{B}} \sum_{i, j \in [n]} \mathbb{E}\left[(\boldsymbol{z}_i \boldsymbol{z}_i^\top \nabla \mathcal{L}(\boldsymbol{\theta}; \{(\boldsymbol{x}_1, \boldsymbol{y}_1)\}))(\boldsymbol{z}_j \boldsymbol{z}_j^\top \nabla \mathcal{L}(\boldsymbol{\theta}; \{(\boldsymbol{x}_2, \boldsymbol{y}_2)\}))^\top\right]$$

Let $\boldsymbol{u}, \boldsymbol{v}$ be two arbitrary vectors. We have that

$$\mathbb{E}_{\boldsymbol{z}_i, \boldsymbol{z}_j}[\boldsymbol{z}_i \boldsymbol{z}_i^\top \boldsymbol{u} \boldsymbol{v}^\top \boldsymbol{z}_j \boldsymbol{z}_j^\top] = \boldsymbol{u} \boldsymbol{v}^\top$$

when $i \neq j$, and

$$\mathbb{E}_{\boldsymbol{z}_i}[\boldsymbol{z}_i \boldsymbol{z}_i^\top \boldsymbol{u} \boldsymbol{v}^\top \boldsymbol{z}_i \boldsymbol{z}_i^\top] = \mathbb{E}_{\boldsymbol{z}}[\boldsymbol{z}^{\otimes 4}](\boldsymbol{u}, \boldsymbol{v})$$
$$= \frac{3d}{d+2} \mathrm{Sym}(\boldsymbol{I}^{\otimes 2})(\boldsymbol{u}, \boldsymbol{v})$$
$$= \frac{d}{d+2} \cdot \boldsymbol{u}^\top \boldsymbol{v} \cdot \boldsymbol{I} + \frac{2d}{d+2} \cdot \boldsymbol{u} \boldsymbol{v}^\top.$$

Therefore

$$\mathbb{E}\left[\widehat{\nabla}\mathcal{L}(\boldsymbol{\theta}; \mathcal{B})\widehat{\nabla}\mathcal{L}(\boldsymbol{\theta}; \mathcal{B})^\top\right]$$
$$= \frac{1}{B^2} \sum_{(\boldsymbol{x}_1, \boldsymbol{y}_1), (\boldsymbol{x}_2, \boldsymbol{y}_2) \in \mathcal{B}} \left(\frac{n-1}{n} + \frac{2d}{n(d+2)}\right) \mathbb{E}\left[\mathcal{L}(\boldsymbol{\theta}; \{(\boldsymbol{x}_1, \boldsymbol{y}_1)\})\mathcal{L}(\boldsymbol{\theta}; \{(\boldsymbol{x}_2, \boldsymbol{y}_2)\})^\top\right]$$
$$+ \frac{d}{n(d+2)} \cdot \mathbb{E}\left[\mathcal{L}(\boldsymbol{\theta}; \{(\boldsymbol{x}_1, \boldsymbol{y}_1)\})^\top \mathcal{L}(\boldsymbol{\theta}; \{(\boldsymbol{x}_2, \boldsymbol{y}_2)\})\right] \boldsymbol{I}.$$

Next, note that when $(\boldsymbol{x}_1, \boldsymbol{y}_1) \neq (\boldsymbol{x}_2, \boldsymbol{y}_2)$, we have

$$\mathbb{E}\left[\mathcal{L}(\boldsymbol{\theta}; \{(\boldsymbol{x}_1, \boldsymbol{y}_1)\})\mathcal{L}(\boldsymbol{\theta}; \{(\boldsymbol{x}_2, \boldsymbol{y}_2)\})^\top\right] = \nabla \mathcal{L}(\boldsymbol{\theta}) \nabla \mathcal{L}(\boldsymbol{\theta})^\top,$$

and when $(\boldsymbol{x}_1, \boldsymbol{y}_1) = (\boldsymbol{x}_2, \boldsymbol{y}_2)$ we have

$$\mathbb{E}\left[\mathcal{L}(\boldsymbol{\theta}; \{(\boldsymbol{x}_1, \boldsymbol{y}_1)\})\mathcal{L}(\boldsymbol{\theta}; \{(\boldsymbol{x}_2, \boldsymbol{y}_2)\})^\top\right] = \nabla \mathcal{L}(\boldsymbol{\theta}) \nabla \mathcal{L}(\boldsymbol{\theta})^\top + \boldsymbol{\Sigma}_{MB}(\boldsymbol{\theta}).$$

Therefore

$$\frac{1}{B^2} \sum_{(\boldsymbol{x}_1, \boldsymbol{y}_1), (\boldsymbol{x}_2, \boldsymbol{y}_2) \in \mathcal{B}} \mathbb{E}\left[\mathcal{L}(\boldsymbol{\theta}; \{(\boldsymbol{x}_1, \boldsymbol{y}_1)\})\mathcal{L}(\boldsymbol{\theta}; \{(\boldsymbol{x}_2, \boldsymbol{y}_2)\})^\top\right] = \nabla \mathcal{L}(\boldsymbol{\theta}) \nabla \mathcal{L}(\boldsymbol{\theta})^\top + \frac{1}{B} \boldsymbol{\Sigma}(\boldsymbol{\theta}),$$

and plugging this yields

$$\mathbb{E}\left[\widehat{\nabla}\mathcal{L}(\boldsymbol{\theta}; \mathcal{B})\widehat{\nabla}\mathcal{L}(\boldsymbol{\theta}; \mathcal{B})^\top\right] = \left(1 + \frac{d-2}{n(d+2)}\right) \cdot \left(\nabla \mathcal{L}(\boldsymbol{\theta}) \nabla \mathcal{L}(\boldsymbol{\theta})^\top + \frac{1}{B} \boldsymbol{\Sigma}(\boldsymbol{\theta})\right)$$
$$+ \frac{d}{n(d+2)} \boldsymbol{I} \cdot \left(\|\nabla \mathcal{L}(\boldsymbol{\theta})\|^2 + \frac{1}{B} \mathrm{tr}(\boldsymbol{\Sigma}(\boldsymbol{\theta}))\right). \tag{7}$$

Finally, we have

$$\mathbb{E}\left[\left\|\widehat{\nabla}\mathcal{L}(\boldsymbol{\theta}; \mathcal{B})\right\|^2\right] = \left(1 + \frac{d^2 + d - 2}{n(d+2)}\right) \cdot \left(\|\nabla \mathcal{L}(\boldsymbol{\theta})\|^2 + \frac{1}{B} \mathrm{tr}(\boldsymbol{\Sigma}(\boldsymbol{\theta}))\right)$$
$$= \frac{d+n-1}{n} \cdot \mathbb{E}\left[\|\nabla \mathcal{L}(\boldsymbol{\theta}; \mathcal{B})\|^2\right].$$

$\square$

*Proof of Theorem 1.* By Taylor's theorem with remainder, we have that

$$\mathcal{L}(\boldsymbol{\theta}_{t+1}) = \mathcal{L}(\boldsymbol{\theta}_t) + \nabla\mathcal{L}(\boldsymbol{\theta}_t)^\top(\boldsymbol{\theta}_{t+1} - \boldsymbol{\theta}_t)$$
$$+ \int_0^1 \lambda(\boldsymbol{\theta}_{t+1} - \boldsymbol{\theta}_t)^\top \nabla^2\mathcal{L}(\lambda\boldsymbol{\theta}_{t+1} + (1-\lambda)\boldsymbol{\theta}_t)(\boldsymbol{\theta}_{t+1} - \boldsymbol{\theta}_t)^\top d\lambda$$

Next, note that

$$\|\boldsymbol{\theta}_{t+1} - \boldsymbol{\theta}_t\| = \eta \left\|\widehat{\nabla}\mathcal{L}(\boldsymbol{\theta};\mathcal{B})\right\| \leq \eta\sqrt{d} \cdot \frac{1}{Bn}\sum|\boldsymbol{z}_i^\top\nabla\mathcal{L}(\boldsymbol{\theta};\{(\boldsymbol{x},\boldsymbol{y})\})| \leq \eta dG(\boldsymbol{\theta}_t).$$

Therefore $\|\lambda\boldsymbol{\theta}_{t+1} + (1-\lambda)\boldsymbol{\theta}_t - \boldsymbol{\theta}_t\| \leq \eta dG(\boldsymbol{\theta}_t)$. By the assumption we have the upper bound $\nabla^2\mathcal{L}(\lambda\boldsymbol{\theta}_{t+1} + (1-\lambda)\boldsymbol{\theta}_t) \preceq \boldsymbol{H}(\boldsymbol{\theta}_t)$, and thus

$$\mathcal{L}(\boldsymbol{\theta}_{t+1}) \leq \mathcal{L}(\boldsymbol{\theta}_t) + \nabla\mathcal{L}(\boldsymbol{\theta}_t)^\top(\boldsymbol{\theta}_{t+1} - \boldsymbol{\theta}_t) + (\boldsymbol{\theta}_{t+1} - \boldsymbol{\theta}_t)^\top\boldsymbol{H}(\boldsymbol{\theta}_t)(\boldsymbol{\theta}_{t+1} - \boldsymbol{\theta}_t)$$
$$= \mathcal{L}(\boldsymbol{\theta}_t) - \eta\nabla\mathcal{L}(\boldsymbol{\theta}_t)^\top\widehat{\nabla}\mathcal{L}(\boldsymbol{\theta}_t;\mathcal{B}) + \frac{1}{2}\eta^2\widehat{\nabla}\mathcal{L}(\boldsymbol{\theta}_t;\mathcal{B})^\top\boldsymbol{H}(\boldsymbol{\theta}_t)\widehat{\nabla}\mathcal{L}(\boldsymbol{\theta}_t;\mathcal{B}).$$

Taking the conditional expectation with respect to $\boldsymbol{\theta}_t$ and plugging in (9), the formula for the covariance of our ZO estimate $\widehat{\nabla}\mathcal{L}(\boldsymbol{\theta}_t;\mathcal{B})$, yields

$$\mathbb{E}[\mathcal{L}(\boldsymbol{\theta}_{t+1}) \mid \boldsymbol{\theta}_t] \leq \mathcal{L}(\boldsymbol{\theta}_t) - \eta\|\nabla\mathcal{L}(\boldsymbol{\theta}_t)\|^2 + \frac{\eta^2}{2}\left\langle\boldsymbol{H}(\boldsymbol{\theta}_t), \mathbb{E}\left[\widehat{\nabla}\mathcal{L}(\boldsymbol{\theta};\mathcal{B})\widehat{\nabla}\mathcal{L}(\boldsymbol{\theta};\mathcal{B})^\top\right]\right\rangle$$
$$= \mathcal{L}(\boldsymbol{\theta}_t) - \eta\|\nabla\mathcal{L}(\boldsymbol{\theta}_t)\|^2 + \frac{\eta^2}{2}\cdot\frac{d}{n(d+2)}\left(\|\nabla\mathcal{L}(\boldsymbol{\theta}_t)\|^2 + \frac{1}{B}\operatorname{tr}(\boldsymbol{\Sigma}(\boldsymbol{\theta}_t))\right)\operatorname{tr}(\boldsymbol{H}(\boldsymbol{\theta}_t))$$
$$+ \frac{\eta^2}{2}\left(1 + \frac{d-2}{n(d+2)}\right)\left(\nabla\mathcal{L}(\boldsymbol{\theta}_t)^\top\boldsymbol{H}(\boldsymbol{\theta}_t)\nabla\mathcal{L}(\boldsymbol{\theta}_t) + \frac{1}{B}\langle\boldsymbol{\Sigma}(\boldsymbol{\theta}_t),\boldsymbol{H}(\boldsymbol{\theta}_t)\rangle\right)$$

By assumption, the Hessian upper bound $\boldsymbol{H}(\boldsymbol{\theta}_t)$ satisfies $\|\boldsymbol{H}(\boldsymbol{\theta}_t)\|_{op} \leq \ell$ and $\operatorname{tr}(\boldsymbol{H}(\boldsymbol{\theta}_t)) \leq \ell r$. Thus

$$\mathbb{E}[\mathcal{L}(\boldsymbol{\theta}_{t+1}) \mid \boldsymbol{\theta}_t] \leq \mathcal{L}(\boldsymbol{\theta}_t) - \eta\|\nabla\mathcal{L}(\boldsymbol{\theta}_t)\|^2 + \frac{\eta^2\ell}{2}\cdot\left(\frac{dr+d-2}{n(d+2)} + 1\right)\cdot\left(\|\nabla\mathcal{L}(\boldsymbol{\theta}_t)\|^2 + \frac{1}{B}\operatorname{tr}(\boldsymbol{\Sigma}(\boldsymbol{\theta}_t))\right)$$
$$= \mathcal{L}(\boldsymbol{\theta}_t) - \eta\|\nabla\mathcal{L}(\boldsymbol{\theta}_t)\|^2 + \frac{\eta^2\ell}{2}\cdot\left(\frac{dr+d-2}{n(d+2)} + 1\right)\cdot\mathbb{E}\left[\|\nabla\mathcal{L}(\boldsymbol{\theta}_t;\mathcal{B})\|^2\right],$$

as desired. $\qquad\square$

## G.1  Proofs of Global Convergence

**Lemma 4.** *Let $\mathcal{L}(\boldsymbol{\theta})$ be $\mu$-PL and let there exist $\alpha$ such that $\operatorname{tr}(\boldsymbol{\Sigma}(\boldsymbol{\theta})) \leq \alpha(\mathcal{L}(\boldsymbol{\theta}) - \mathcal{L}^*)$ for all $\boldsymbol{\theta}$. Then after*

$$t = O\left(\left(\frac{\ell}{\mu} + \frac{\ell\alpha}{\mu^2 B}\right)\log\frac{\mathcal{L}(\boldsymbol{\theta}_0) - \mathcal{L}^*}{\epsilon}\right)$$

*iterations of SGD we have $\mathbb{E}[\mathcal{L}(\boldsymbol{\theta}_t)] \leq \mathcal{L}^* + \epsilon$.*

*Proof of Lemma 4.* The descent lemma for SGD yields

$$\mathbb{E}[\mathcal{L}(\boldsymbol{\theta}_{t+1}) \mid \boldsymbol{\theta}_t] - \mathcal{L}(\boldsymbol{\theta}_t) \leq -\eta\|\nabla\mathcal{L}(\boldsymbol{\theta}_t)\|^2 + \frac{1}{2}\eta^2\ell\cdot\mathbb{E}[\|\nabla\mathcal{L}(\boldsymbol{\theta}_t;\mathcal{B})\|^2].$$

Plugging in $\mathbb{E}[\|\nabla\mathcal{L}(\boldsymbol{\theta}_t;\mathcal{B})\|^2] = \|\nabla\mathcal{L}(\boldsymbol{\theta}_t)\|^2 + \frac{1}{B}\operatorname{tr}(\boldsymbol{\Sigma}(\boldsymbol{\theta}_t))$ and selecting a learning rate $\eta \leq \frac{1}{\ell}$ yields

$$\mathbb{E}[\mathcal{L}(\boldsymbol{\theta}_{t+1}) \mid \boldsymbol{\theta}_t] \leq \mathcal{L}(\boldsymbol{\theta}_t) - \frac{\eta}{2}\|\nabla\mathcal{L}(\boldsymbol{\theta}_t)\|^2 + \frac{\eta^2\ell}{2B}\operatorname{tr}(\boldsymbol{\Sigma}(\boldsymbol{\theta}_t))$$

Since $\mathcal{L}$ is $\mu$-PL, we get

$$\mathbb{E}[\mathcal{L}(\boldsymbol{\theta}_{t+1}) \mid \boldsymbol{\theta}_t] \leq \mathcal{L}(\boldsymbol{\theta}_t) - \eta\mu(\mathcal{L}(\boldsymbol{\theta}_t) - \mathcal{L}^*) + \frac{\eta^2\ell}{2B}\operatorname{tr}(\boldsymbol{\Sigma}(\boldsymbol{\theta}_t)).$$

Since $\text{tr}(\boldsymbol{\Sigma}(\boldsymbol{\theta}_t)) \leq \alpha(\mathcal{L}(\boldsymbol{\theta}_t) - \mathcal{L}^*)$, we have

$$\mathbb{E}[\mathcal{L}(\boldsymbol{\theta}_{t+1}) \mid \boldsymbol{\theta}_t] \leq \mathcal{L}(\boldsymbol{\theta}_t) - \eta\mu(\mathcal{L}(\boldsymbol{\theta}_t) - \mathcal{L}^*) + \frac{\eta^2\ell\alpha}{2B}(\mathcal{L}(\boldsymbol{\theta}_t) - \mathcal{L}^*).$$

Altogether,

$$\mathbb{E}[\mathcal{L}(\boldsymbol{\theta}_{t+1})] - \mathcal{L}^* \leq \left(1 - \eta\mu + \frac{\eta^2\ell\alpha}{2B}\right)(\mathbb{E}[\mathcal{L}(\boldsymbol{\theta}_t)] - \mathcal{L}^*)$$

Choosing $\eta = \min(\frac{1}{\ell}, \frac{\mu B}{\ell\alpha})$, we obtain

$$\mathbb{E}[\mathcal{L}(\boldsymbol{\theta}_{t+1})] - \mathcal{L}^* \leq \left(1 - \min(\frac{\mu}{2\ell}, \frac{\mu^2 B}{2\ell\alpha})\right)(\mathbb{E}[\mathcal{L}(\boldsymbol{\theta}_t)] - \mathcal{L}^*).$$

Therefore we reach a solution with $\mathbb{E}[\mathcal{L}(\boldsymbol{\theta}_t)] - \mathcal{L}^* \leq \epsilon$ after

$$t = \max\left(\frac{2\ell}{\mu}, \frac{2\ell\alpha}{\mu^2 B}\right)\log\left(\frac{\mathcal{L}(\boldsymbol{\theta}_0) - \mathcal{L}^*}{\epsilon}\right) = O\left(\left(\frac{\ell}{\mu} + \frac{\ell\alpha}{\mu^2 B}\right)\log\frac{\mathcal{L}(\boldsymbol{\theta}_0) - \mathcal{L}^*}{\epsilon}\right)$$

iterations. $\qquad\square$

*Proof of Lemma 3.* By Corollary 1, ZO-SGD with $\eta_{\text{ZO}} = \gamma^{-1}\eta_{\text{SGD}}$ yields

$$\mathbb{E}[\mathcal{L}(\boldsymbol{\theta}_{t+1}) \mid \boldsymbol{\theta}_t] - \mathcal{L}(\boldsymbol{\theta}_t) \leq \frac{1}{\gamma} \cdot \left[-\eta_{\text{SGD}}\|\nabla\mathcal{L}(\boldsymbol{\theta}_t)\|^2 + \frac{1}{2}\eta_{\text{SGD}}^2\ell \cdot \mathbb{E}[\|\nabla\mathcal{L}(\boldsymbol{\theta}; \mathcal{B})\|^2]\right].$$

As in the proof for SGD, choosing $\eta_{\text{SGD}} \leq \frac{1}{\ell}$ yields

$$\mathbb{E}[\mathcal{L}(\boldsymbol{\theta}_{t+1}) \mid \boldsymbol{\theta}_t] - \mathcal{L}(\boldsymbol{\theta}_t) \leq \gamma^{-1} \cdot \left[-\frac{\eta_{\text{SGD}}}{2}\|\nabla\mathcal{L}(\boldsymbol{\theta}_t)\|^2 + \frac{\eta_{\text{SGD}}^2\ell}{2B}\text{tr}(\boldsymbol{\Sigma}(\boldsymbol{\theta}_t))\right].$$

Therefore under $\mu$-PL and the $\text{tr}(\boldsymbol{\Sigma}(\boldsymbol{\theta}_t)) \leq \alpha(\mathcal{L}(\boldsymbol{\theta}_t) - \mathcal{L}^*)$ assumption we obtain

$$\mathbb{E}[\mathcal{L}(\boldsymbol{\theta}_{t+1})] - \mathbb{E}[\mathcal{L}(\boldsymbol{\theta}_t)] \leq \gamma^{-1} \cdot \left[-\eta_{\text{SGD}}\mu + \frac{\eta_{\text{SGD}}^2\ell\alpha}{2B}\right] \cdot (\mathbb{E}[\mathcal{L}(\boldsymbol{\theta}_t)] - \mathcal{L}^*)$$

$$\implies \mathbb{E}[\mathcal{L}(\boldsymbol{\theta}_{t+1})] - \mathcal{L}^* \leq \left(1 - \gamma^{-1}\left(\eta_{\text{SGD}}\mu - \frac{\eta_{\text{SGD}}^2\ell\alpha}{2B}\right)\right)(\mathbb{E}[\mathcal{L}(\boldsymbol{\theta}_t)] - \mathcal{L}^*).$$

Choosing $\eta_{\text{SGD}} = \min(\frac{1}{\ell}, \frac{\mu B}{\ell\alpha})$ yields

$$\mathbb{E}[\mathcal{L}(\boldsymbol{\theta}_{t+1})] - \mathcal{L}^* \leq \left(1 - \gamma^{-1} \cdot \min(\frac{\mu}{2\ell}, \frac{\mu^2 B}{2\ell\alpha})\right)(\mathbb{E}[\mathcal{L}(\boldsymbol{\theta}_t)] - \mathcal{L}^*).$$

Therefore we reach a solution with $\mathbb{E}[\mathcal{L}(\boldsymbol{\theta}_t)] - \mathcal{L}^* \leq \epsilon$ after

$$t = \gamma \cdot \max\left(\frac{2\ell}{\mu}, \frac{2\ell\alpha}{\mu^2 B}\right)\log\left(\frac{\mathcal{L}(\boldsymbol{\theta}_0) - \mathcal{L}^*}{\epsilon}\right) = \mathcal{O}\left(\left(\frac{r}{n} + 1\right) \cdot \left(\frac{\ell}{\mu} + \frac{\ell\alpha}{\mu^2 B}\right)\log\frac{\mathcal{L}(\boldsymbol{\theta}_0) - \mathcal{L}^*}{\epsilon}\right)$$

iterations. $\qquad\square$

### G.1.1 Verification of assumptions

We show that the $\text{tr}(\boldsymbol{\Sigma}(\boldsymbol{\theta}_t)) \leq \alpha(\mathcal{L}(\boldsymbol{\theta}_t) - \mathcal{L}^*)$ assumption holds for certain losses.

First, consider optimizing the model $f(\boldsymbol{x}; \boldsymbol{\theta})$ with square loss, so that

$$\mathcal{L}(\boldsymbol{\theta}) = \frac{1}{N}\sum_{i \in [N]}(f(\boldsymbol{x}_i; \boldsymbol{\theta}) - \boldsymbol{y}_i)^2.$$

One then has that

$$\boldsymbol{\Sigma}(\boldsymbol{\theta}) = \frac{2}{N}\sum_{i \in [N]}(f(\boldsymbol{x}_i; \boldsymbol{\theta}) - \boldsymbol{y}_i)^2\nabla f(\boldsymbol{x}_i; \boldsymbol{\theta})\nabla f(\boldsymbol{x}_i; \boldsymbol{\theta})^\top - \nabla\mathcal{L}(\boldsymbol{\theta})\nabla\mathcal{L}(\boldsymbol{\theta})^\top.$$

Therefore

$$\text{tr}(\mathbf{\Sigma}(\boldsymbol{\theta})) \le \frac{2}{N} \sum_{i \in [N]} (f(\boldsymbol{x}_i; \boldsymbol{\theta}) - \boldsymbol{y}_i)^2 \|\nabla f(\boldsymbol{x}_i; \boldsymbol{\theta})\|^2$$

$$\le 2\mathcal{L}(\boldsymbol{\theta}) \sum_{i \in [N]} \|\nabla f(\boldsymbol{x}_i; \boldsymbol{\theta})\|^2 .$$

Assume that the data can be interpolated, i.e., $\mathcal{L}^* = 0$. If the function is $L$-Lipschitz, i.e., $\|\nabla f(\boldsymbol{x}; \boldsymbol{\theta})\| \le L$, then the condition holds with $\alpha = 2NL^2$. If we are in the kernel regime, i.e., $f(\boldsymbol{x}_i; \boldsymbol{\theta}) = \phi(\boldsymbol{x}_i)^\top \boldsymbol{\theta}$ for some feature map $\phi$, then

$$\nabla^2 \mathcal{L}(\boldsymbol{\theta}) = \frac{2}{N} \sum_{i \in [N]} f(\boldsymbol{x}_i; \boldsymbol{\theta}) \nabla f(\boldsymbol{x}_i; \boldsymbol{\theta})^\top .$$

Thus

$$\text{tr}(\mathbf{\Sigma}(\boldsymbol{\theta})) \le N \, \text{tr}(\nabla^2 \mathcal{L}(\boldsymbol{\theta})) \cdot \mathcal{L}(\boldsymbol{\theta}) \le N\ell r \cdot \mathcal{L}(\boldsymbol{\theta}).$$

So the condition holds for $\alpha = N\ell r$.

Next, consider the cross entropy loss function, i.e

$$\mathcal{L}(\boldsymbol{\theta}) = \frac{1}{N} \sum_{i \in [N]} \exp(-y_i f(\boldsymbol{x}_i; \boldsymbol{\theta})).$$

One then has that

$$\mathbf{\Sigma}(\boldsymbol{\theta}) = \frac{1}{N} \sum_{i \in [N]} \exp(-2y_i f(\boldsymbol{x}_i; \boldsymbol{\theta})) y_i^2 \nabla f(\boldsymbol{x}_i; \boldsymbol{\theta}) \nabla f(\boldsymbol{x}_i; \boldsymbol{\theta})^\top - \mathcal{L}(\boldsymbol{\theta})\mathcal{L}(\boldsymbol{\theta})^\top,$$

Assume that the targets $y_i$ are bounded in $[-1, 1]$ (which is true for binary classification tasks), and that $\mathcal{L}^* = 0$ (which can be achieved if $|f(\boldsymbol{x}; \boldsymbol{\theta})|$ can be sent to $\infty$) we have that

$$\text{tr}(\mathbf{\Sigma}(\boldsymbol{\theta})) \le \frac{1}{N} \sum_{i \in [N]} \exp(-2y_i f(\boldsymbol{x}_i; \boldsymbol{\theta})) \|\nabla f(\boldsymbol{x}_i; \boldsymbol{\theta})\|^2 .$$

In the kernel regime, $f(\boldsymbol{x}_i; \boldsymbol{\theta}) = \phi(\boldsymbol{x}_i)^\top \boldsymbol{\theta}$, and thus

$$\nabla^2 \mathcal{L}(\boldsymbol{\theta}) = \frac{1}{N} \sum_{i \in [N]} \exp(-y_i f(\boldsymbol{x}_i; \boldsymbol{\theta})) \nabla f(\boldsymbol{x}_i; \boldsymbol{\theta}) \nabla f(\boldsymbol{x}_i; \boldsymbol{\theta})^\top .$$

Therefore

$$\text{tr}(\mathbf{\Sigma}(\boldsymbol{\theta})) \le N \, \text{tr}(\nabla^2 \mathcal{L}(\boldsymbol{\theta})) \cdot \mathcal{L}(\boldsymbol{\theta}) \le N\ell r \cdot \mathcal{L}(\boldsymbol{\theta}).$$

Therefore the condition holds with $\alpha = N\ell r$ as well.

## G.2 Proofs for Gaussian perturbations

The first lemma computes the second moment of the covariance estimate $\widehat{\nabla}\mathcal{L}(\boldsymbol{\theta}; \mathcal{B})$ when $\boldsymbol{z}$ is drawn $\mathcal{N}(0, \boldsymbol{I})$.

**Lemma 5.** *Let $\boldsymbol{z}_i \sim \mathcal{N}(0, \boldsymbol{I})$ i.i.d. Then*

$$\mathbb{E}\left[\widehat{\nabla}\mathcal{L}(\boldsymbol{\theta}; \mathcal{B})\widehat{\nabla}\mathcal{L}(\boldsymbol{\theta}; \mathcal{B})^\top\right] = \left(1 + \frac{1}{n}\right) \cdot \left(\nabla\mathcal{L}(\boldsymbol{\theta})\nabla\mathcal{L}(\boldsymbol{\theta})^\top + \frac{1}{B}\mathbf{\Sigma}_{MB}(\boldsymbol{\theta})\right)$$

$$+ \frac{1}{n}\boldsymbol{I} \cdot \left(\|\nabla\mathcal{L}(\boldsymbol{\theta})\|^2 + \frac{1}{B}\text{tr}(\mathbf{\Sigma}_{MB}(\boldsymbol{\theta}))\right). \tag{8}$$

*Proof.* As in the proof of Lemma 2, we have that in the $\epsilon \to 0$ limit

$$\mathbb{E}\left[\widehat{\nabla}\mathcal{L}(\boldsymbol{\theta}; \mathcal{B})\widehat{\nabla}\mathcal{L}(\boldsymbol{\theta}; \mathcal{B})^\top\right]$$

$$= \frac{1}{B^2 n^2} \sum_{(\boldsymbol{x}_1, \boldsymbol{y}_1), (\boldsymbol{x}_2, \boldsymbol{y}_2) \in \mathcal{B}} \sum_{i,j \in [n]} \mathbb{E}\left[(\boldsymbol{z}_i \boldsymbol{z}_i^\top \nabla\mathcal{L}(\boldsymbol{\theta}; \{(\boldsymbol{x}_1, \boldsymbol{y}_1)\}))(\boldsymbol{z}_j \boldsymbol{z}_j^\top \nabla\mathcal{L}(\boldsymbol{\theta}; \{(\boldsymbol{x}_2, \boldsymbol{y}_2)\}))^\top\right]$$

For vectors $\boldsymbol{u}, \boldsymbol{v}$, we have that

$$\mathbb{E}_{\boldsymbol{z}_i, \boldsymbol{z}_j}[\boldsymbol{z}_i \boldsymbol{z}_i^\top \boldsymbol{u} \boldsymbol{v}^\top \boldsymbol{z}_j \boldsymbol{z}_j^\top] = \boldsymbol{u} \boldsymbol{v}^\top$$

when $i \neq j$, and

$$\mathbb{E}_{\boldsymbol{z}_i}[\boldsymbol{z}_i \boldsymbol{z}_i^\top \boldsymbol{u} \boldsymbol{v}^\top \boldsymbol{z}_i \boldsymbol{z}_i^\top] = \mathbb{E}_{\boldsymbol{z}}[\boldsymbol{z}^{\otimes 4}](\boldsymbol{u}, \boldsymbol{v}) = 3\mathrm{Sym}(\boldsymbol{I}^{\otimes 2})(\boldsymbol{u}, \boldsymbol{v}) = \boldsymbol{u}^\top \boldsymbol{v} \cdot \boldsymbol{I} + 2\boldsymbol{u} \boldsymbol{v}^\top.$$

Therefore

$$
\begin{aligned}
&\mathbb{E}\left[\widehat{\nabla}\mathcal{L}(\boldsymbol{\theta}; \mathcal{B})\widehat{\nabla}\mathcal{L}(\boldsymbol{\theta}; \mathcal{B})^\top\right] \\
&= \frac{1}{B^2} \sum_{(\boldsymbol{x}_1, \boldsymbol{y}_1), (\boldsymbol{x}_2, \boldsymbol{y}_2) \in \mathcal{B}} \left(\frac{n-1}{n} + \frac{2}{n}\right) \mathbb{E}\left[\mathcal{L}(\boldsymbol{\theta}; \{(\boldsymbol{x}_1, \boldsymbol{y}_1)\})\mathcal{L}(\boldsymbol{\theta}; \{(\boldsymbol{x}_2, \boldsymbol{y}_2)\})^\top\right] \\
&\quad + \frac{1}{n} \cdot \mathbb{E}\left[\mathcal{L}(\boldsymbol{\theta}; \{(\boldsymbol{x}_1, \boldsymbol{y}_1)\})^\top \mathcal{L}(\boldsymbol{\theta}; \{(\boldsymbol{x}_2, \boldsymbol{y}_2)\})\right] \boldsymbol{I}.
\end{aligned}
$$

In the proof of Lemma 2 we showed that

$$\frac{1}{B^2} \sum_{(\boldsymbol{x}_1, \boldsymbol{y}_1), (\boldsymbol{x}_2, \boldsymbol{y}_2) \in \mathcal{B}} \mathbb{E}\left[\mathcal{L}(\boldsymbol{\theta}; \{(\boldsymbol{x}_1, \boldsymbol{y}_1)\})\mathcal{L}(\boldsymbol{\theta}; \{(\boldsymbol{x}_2, \boldsymbol{y}_2)\})^\top\right] = \nabla\mathcal{L}(\boldsymbol{\theta})\nabla\mathcal{L}(\boldsymbol{\theta})^\top + \frac{1}{B}\boldsymbol{\Sigma}(\boldsymbol{\theta}).$$

Plugging this yields

$$
\begin{aligned}
\mathbb{E}\left[\widehat{\nabla}\mathcal{L}(\boldsymbol{\theta}; \mathcal{B})\widehat{\nabla}\mathcal{L}(\boldsymbol{\theta}; \mathcal{B})^\top\right] &= \left(\frac{n+1}{n}\right) \cdot \left(\nabla\mathcal{L}(\boldsymbol{\theta})\nabla\mathcal{L}(\boldsymbol{\theta})^\top + \frac{1}{B}\boldsymbol{\Sigma}(\boldsymbol{\theta})\right) \\
&\quad + \frac{1}{n}\boldsymbol{I} \cdot \left(\|\nabla\mathcal{L}(\boldsymbol{\theta})\|^2 + \frac{1}{B}\mathrm{tr}(\boldsymbol{\Sigma}(\boldsymbol{\theta}))\right).
\end{aligned}
\tag{9}
$$

$\square$

We can prove an analog to Theorem 1 in the case where the $\boldsymbol{z}_i$ are Gaussian. One challenge is that $\|\boldsymbol{\theta}_{t+1} - \boldsymbol{\theta}_t\|$ is no longer bounded; instead we the $r$-local effective rank assumption only holds with high probability, and thus to bound the expected loss decrease we must control the probability of the $\|\boldsymbol{\theta}_{t+1} - \boldsymbol{\theta}_t\|$ being large.

Consider the following modified version of the local $r$-effective rank assumption, where the upper bound on the Hessian is measured over a ball of radius twice as large as the one in Assumption 1.

**Assumption 2** (Local $r$-effective rank, Gaussian). *Let $G(\boldsymbol{\theta}_t) = \max_{(\boldsymbol{x}, \boldsymbol{y}) \in \mathcal{D}} \|\nabla\mathcal{L}(\boldsymbol{\theta}_t; \{(\boldsymbol{x}, \boldsymbol{y})\})\|$. There exists a matrix $\boldsymbol{H}(\boldsymbol{\theta}_t)$ such that:*

1. *For all $\boldsymbol{\theta}$ such that $\|\boldsymbol{\theta} - \boldsymbol{\theta}_t\| \leq 2\eta d G(\boldsymbol{\theta}_t)$, we have $\nabla^2\mathcal{L}(\boldsymbol{\theta}) \preceq \boldsymbol{H}(\boldsymbol{\theta}_t)$.*

2. *The effective rank of $\boldsymbol{H}(\boldsymbol{\theta}_t)$, i.e., $\mathrm{tr}(\boldsymbol{H}(\boldsymbol{\theta}_t))/\|\boldsymbol{H}(\boldsymbol{\theta}_t)\|_{op}$, is at most $r$.*

**Theorem 2** (Dimension-Free Rate, Gaussian $\boldsymbol{z}$). *Assume the loss exhibits local $r$-effective rank (Assumption 2). If $\boldsymbol{\theta}_{t+1} = \boldsymbol{\theta}_t - \eta_{\mathrm{ZO}}\widehat{\nabla}\mathcal{L}(\boldsymbol{\theta}_t; \mathcal{B})$ is a single step of ZO-SGD using the $n$-SPSA estimate with a minibatch of size $B$, then there exists a $\gamma = \Theta(r/n)$ such the expected loss decrease can be bounded as*

$$
\begin{aligned}
&\mathbb{E}[\mathcal{L}(\boldsymbol{\theta}_{t+1}) \mid \boldsymbol{\theta}_t] - \mathcal{L}(\boldsymbol{\theta}_t) \\
&\leq -\eta_{\mathrm{ZO}}\|\nabla\mathcal{L}(\boldsymbol{\theta}_t)\|^2 + \frac{1}{2}\eta_{\mathrm{ZO}}^2 \ell \cdot \gamma \cdot \mathbb{E}[\|\nabla\mathcal{L}(\boldsymbol{\theta}_t; \mathcal{B})\|^2] + \eta_{\mathrm{ZO}}^2 \ell G(\boldsymbol{\theta}_t)^2 \exp(-\Omega(nd)).
\end{aligned}
$$

*Proof of Theorem 2.* Let $\mathcal{A}$ be the event that $\|\boldsymbol{\theta}_{t+1} - \boldsymbol{\theta}_t\| \leq 2\eta d G(\boldsymbol{\theta}_t)$. On $\mathcal{A}$, we have that

$$\mathcal{L}(\boldsymbol{\theta}_{t+1}) \leq \mathcal{L}(\boldsymbol{\theta}_t) - \eta\nabla\mathcal{L}(\boldsymbol{\theta}_t)^\top \widehat{\nabla}\mathcal{L}(\boldsymbol{\theta}; \mathcal{B}) + \frac{1}{2}\eta^2 \widehat{\nabla}\mathcal{L}(\boldsymbol{\theta}_t; \mathcal{B})^\top \boldsymbol{H}(\boldsymbol{\theta})\widehat{\nabla}\mathcal{L}(\boldsymbol{\theta}_t; \mathcal{B}).$$

Likewise, since $\mathcal{L}$ is $\ell$-smooth, we have that

$$\mathcal{L}(\boldsymbol{\theta}_{t+1}) \leq \mathcal{L}(\boldsymbol{\theta}_t) - \eta\nabla\mathcal{L}(\boldsymbol{\theta}_t)^\top \widehat{\nabla}\mathcal{L}(\boldsymbol{\theta}; \mathcal{B}) + \frac{1}{2}\eta^2 \ell \left\|\widehat{\nabla}\mathcal{L}(\boldsymbol{\theta}_t; \mathcal{B})\right\|^2.$$

Therefore

$$\mathbb{E}[\mathcal{L}(\boldsymbol{\theta}_{t+1}) \mid \boldsymbol{\theta}_t] \le \mathcal{L}(\boldsymbol{\theta}_{t+1}) - \eta \|\nabla\mathcal{L}(\boldsymbol{\theta}_t)\|^2 + \frac{1}{2}\eta^2 \left\langle \mathbb{E}\left[\widehat{\nabla}\mathcal{L}(\boldsymbol{\theta};\mathcal{B})\widehat{\nabla}\mathcal{L}(\boldsymbol{\theta};\mathcal{B})^\top \cdot \mathbf{1}(\mathcal{A})\right], \boldsymbol{H}(\boldsymbol{\theta}_t)\right\rangle$$
$$+ \frac{1}{2}\eta^2\ell\mathbb{E}\left[\left\|\widehat{\nabla}\mathcal{L}(\boldsymbol{\theta}_t;\mathcal{B})\right\|^2 \cdot \mathbf{1}(\neg\mathcal{A})\right]$$
$$= \mathcal{L}(\boldsymbol{\theta}_{t+1}) - \eta\|\nabla\mathcal{L}(\boldsymbol{\theta}_t)\|^2 + \frac{1}{2}\eta^2 \left\langle \mathbb{E}\left[\widehat{\nabla}\mathcal{L}(\boldsymbol{\theta};\mathcal{B})\widehat{\nabla}\mathcal{L}(\boldsymbol{\theta};\mathcal{B})^\top\right], \boldsymbol{H}(\boldsymbol{\theta}_t)\right\rangle$$
$$\frac{1}{2}\eta^2 \left\langle \mathbb{E}\left[\widehat{\nabla}\mathcal{L}(\boldsymbol{\theta};\mathcal{B})\widehat{\nabla}\mathcal{L}(\boldsymbol{\theta};\mathcal{B})^\top \cdot \mathbf{1}(\neg\mathcal{A})\right], \ell I - \boldsymbol{H}(\boldsymbol{\theta}_t)\right\rangle.$$

The latter term can be bounded as follows

$$\frac{1}{2}\eta^2 \left\langle \mathbb{E}\left[\widehat{\nabla}\mathcal{L}(\boldsymbol{\theta};\mathcal{B})\widehat{\nabla}\mathcal{L}(\boldsymbol{\theta};\mathcal{B})^\top \cdot \mathbf{1}(\neg\mathcal{A})\right], \ell I - \boldsymbol{H}(\boldsymbol{\theta}_t)\right\rangle \le \eta^2\ell\mathbb{E}\left[\left\|\widehat{\nabla}\mathcal{L}(\boldsymbol{\theta};\mathcal{B})\right\|^2 \cdot \mathbf{1}(\neg\mathcal{A})\right]$$
$$\le \eta^2\ell\mathbb{E}\left[\left\|\widehat{\nabla}\mathcal{L}(\boldsymbol{\theta};\mathcal{B})\right\|^4\right]^{\frac{1}{2}} \Pr[\neg\mathcal{A}]^{1/2}.$$

The gradient estimate $\widehat{\nabla}\mathcal{L}(\boldsymbol{\theta};\mathcal{B})$ satisfies

$$\left\|\widehat{\nabla}\mathcal{L}(\boldsymbol{\theta};\mathcal{B})\right\| \le \frac{1}{n}\sum_{i\in[n]} |\boldsymbol{z}_i^\top\nabla\mathcal{L}(\boldsymbol{\theta};\mathcal{B})| \cdot \|\boldsymbol{z}_i\|$$

The expectation term is upper bounded as

$$\mathbb{E}\left[\left\|\widehat{\nabla}\mathcal{L}(\boldsymbol{\theta};\mathcal{B})\right\|^4\right] \le \frac{1}{n}\sum_{i\in[n]} \mathbb{E}\left[|\boldsymbol{z}^\top\nabla\mathcal{L}(\boldsymbol{\theta};\mathcal{B})|^4 \cdot \|\boldsymbol{z}\|^4\right]$$
$$\le \mathbb{E}\left[|\boldsymbol{z}^\top\nabla\mathcal{L}(\boldsymbol{\theta};\mathcal{B})|^8\right]^{1/2} \mathbb{E}\left[\|\boldsymbol{z}\|^8\right]^{1/2}$$
$$\le \sqrt{105}(d+6)^2 G(\boldsymbol{\theta}_t)^4,$$

where we have plugged in explicit formulas for moments of Gaussian and $\chi^2$ random variables. Next, note that on the event $\neg\mathcal{A}$, we have

$$2\eta d G(\boldsymbol{\theta}_t) \le \|\boldsymbol{\theta}_{t+1} - \boldsymbol{\theta}_t\| = \eta \left\|\widehat{\nabla}\mathcal{L}(\boldsymbol{\theta}_t;\mathcal{B})\right\| \le \eta \cdot \frac{1}{n}\sum_{i\in[n]} \|\boldsymbol{z}_i\|^2 G(\boldsymbol{\theta}_t).$$

Therefore

$$\Pr[\neg\mathcal{A}] \le \Pr\left[\sum_{i\in[n]} \|\boldsymbol{z}_i\|^2 \ge 2nd\right]$$

**Lemma 6** (Standard $\chi^2$-tail bound). *Let Z be a $\chi^2$ random variable with k degrees of freedom. Then*

$$\Pr[Z \ge k + u] \le \exp\left(-\min\left(\frac{u^2}{16k}, \frac{u}{16}\right)\right)$$

Since $\sum_{i\in[n]} \|\boldsymbol{z}_i\|^2$ is a $\chi^2$ random variable with $nd$ degrees of freedom, we thus have that

$$\Pr[\neg\mathcal{A}] \le \exp\left(-\frac{nd}{16}\right).$$

Altogether,

$$\frac{1}{2}\eta^2 \left\langle \mathbb{E}\left[\widehat{\nabla}\mathcal{L}(\boldsymbol{\theta};\mathcal{B})\widehat{\nabla}\mathcal{L}(\boldsymbol{\theta};\mathcal{B})^\top \cdot \mathbf{1}(\neg\mathcal{A})\right], \ell I - \boldsymbol{H}(\boldsymbol{\theta}_t)\right\rangle \le \eta^2\ell 105^{1/4}(d+6)G(\boldsymbol{\theta}_t)^2 \exp(-\frac{nd}{32})$$
$$= \eta^2\ell G(\boldsymbol{\theta}_t)^2 \exp(-\Omega(nd)).$$

Finally, plugging in (8), along with the fact that $\|\boldsymbol{H}(\boldsymbol{\theta}_t)\|_{op} \leq \ell$ and $\mathrm{tr}(\boldsymbol{H}(\boldsymbol{\theta}_t)) \leq \ell r$,

$$\left\langle \mathbb{E}\left[\widehat{\nabla}\mathcal{L}(\boldsymbol{\theta};\mathcal{B})\widehat{\nabla}\mathcal{L}(\boldsymbol{\theta};\mathcal{B})^\top\right], \boldsymbol{H}(\boldsymbol{\theta}_t)\right\rangle = \frac{r+n+1}{n} \cdot \ell\left(\|\nabla\mathcal{L}(\boldsymbol{\theta}_t)\|^2 + \frac{1}{B}\mathrm{tr}(\boldsymbol{\Sigma}(\boldsymbol{\theta}_t))\right)$$

$$= \frac{r+n+1}{n} \cdot \mathbb{E}\left[\|\nabla\mathcal{L}(\boldsymbol{\theta}_t;\mathcal{B})\|^2\right]$$

Thus letting $\gamma = \frac{r+n+1}{n}$ yields

$$\mathbb{E}[\mathcal{L}(\boldsymbol{\theta}_{t+1}) \mid \boldsymbol{\theta}_t] - \mathcal{L}(\boldsymbol{\theta}_t)$$

$$\leq -\eta\|\nabla\mathcal{L}(\boldsymbol{\theta}_t)\|^2 + \frac{1}{2}\eta^2\ell \cdot \gamma \cdot \mathbb{E}[\|\nabla\mathcal{L}(\boldsymbol{\theta}_t;\mathcal{B})\|^2] + \eta^2\ell G(\boldsymbol{\theta}_t)^2 \exp(-\Omega(nd)),$$

as desired. $\qquad\square$

