# OpenReview forum: "Fine-Tuning Language Models with Just Forward Passes"
_NeurIPS.cc/2023/Conference — NeurIPS 2023 oral_

### Official Review · Reviewer_XvSx · 2023-07-04

**Soundness:** 4 excellent
**Presentation:** 4 excellent
**Contribution:** 4 excellent
**Rating:** 7
**Confidence:** 4

**Summary:**

Fine-tuning with backpropogation becomes infeasible for very large language models because it uses too much memory. While zeroth-order optimization uses far less memory and could in principle fine-tune the model with just forward passes, past theory suggested that the learning rate must scale down with the number of parameters, making convergence prohibitively slow. However, this paper finds that zeroth-order optimization actually performs quite well and converges quickly even on very large language models. They provide theory to explain this fast convergence, where they show that under an assumption they call "low effective rank," the learning rate scales down with the rank rather than the number of parameters. They also provide a memory-efficient implementation of zeroth-order optimization that they call MeZO, along with memory efficient zeroth-order versions of SGD with momentum and Adam. In experiments, the method performs similarly to backpropogation with 1/12 the memory usage, while outperforming in-context learning and linear probing.

**Strengths:**

(1) The paper is well-written.

(2) The method is simple and easy to understand.

(3) The theory provides useful insights into why zeroth-order optimization works for fine-tuning large pre-trained models.

(4) The experimental results are strong, and the appendix contains thorough ablations.

(5) The idea that zeroth-order optimization works well for fine-tuning LMs seems practically useful and addresses a pressing need in the community for memory-efficient methods.

**Weaknesses:**

The paper seems strong overall, and I support its acceptance regardless of whether the suggested experiments below are run or not during the rebuttal period.

(1) From what I understand, the paper does not verify the low effective rank assumption empirically, nor is it verified in the papers cited (which either study the effective rank / Hessian spectra in non-LLMs, or study the LLMs but not the effective rank and instead study the intrinsic dimensionality of fine-tuning). Therefore, to justify the assumption, it seems useful to study the Hessian spectra of the downstream fine-tuning loss in LLMs, at whatever size is feasible.

(2) Related to (1), to verify the theory and confirm that the effective rank is indeed the quantity that determines convergence rates, it seems useful to run simulated experiments where one constructs a synthetic model + data and varies the effective rank, and examines whether the convergence rate or gradient norm scales with the effective rank as predicted in the theory.

**Questions:**

(1) While MeZO is much more memory efficient, is it slower than backprop in terms of wall-clock time? (The appendix does state that FT used 1K steps while MeZO used 100K steps in the experiments. How much faster is each step of MeZO compared to each step of backprop?)

(2) It's a bit surprising to me that MeZO, MeZO-prefix, and MeZO-LoRA optimize at similar speeds (Figure 5 in the appendix). Do the backprop versions also optimize at similar speeds? And do the three parameterizations actually have similar effective ranks empirically?

---

> ### Author Rebuttal · Authors · 2023-08-10
>
> **Can we study the effective rank assumption in some language models?**
>
> It is difficult to translate results on very small models, on which we would be able to measure the effective rank, to the large ones that we would find MeZO useful for. We would also likely need to pre-train these very small models ourselves, which is expensive, and they may be so small that they do not achieve meaningful results on the benchmarks we study.
>
> **Can we use simulations to verify the theoretical convergence analysis?**
>
> Thanks for the suggestion! We added the simulated experiment in our attached PDF and also reported the results in our general response. In short, we observed that the convergence rate of MeZO does depend on the effective rank in the simulated experiments.
>
>
> **What is the wall-clock efficiency of MeZO compared to fine-tuning with backpropagation?**
>
> Please see our general response. In short, MeZO reduces the number of GPU-hours needed to train large models, leading to a 2x GPU-hour reduction on a 30B model compared to Adam fine-tuning.
>
> **Do the backpropagation versions of full fine-tuning, prefix tuning, and LoRA optimize at roughly the same rates, as was observed with MeZO? And do the three parameterizations actually have similar effective ranks empirically?**
>
> We did not measure the empirical effective rank of different methods due to limited compute. We observe that with backpropagation, all three methods converge roughly at a similar speed, with LoRA and prefix-tuning slightly slower on some tasks. The interesting case with MeZO is that classical ZO analyses suggest that full-parameter MeZO would converge much more slowly, but it is not the case empirically. Our theory in Section 4 highlights why the convergence rate does not depend on the number of parameters.

---

> > ### Comment · Reviewer_XvSx · 2023-08-15
> >
> > Thanks for the great answers!

---

### Official Review · Reviewer_Qhm1 · 2023-07-06

**Soundness:** 3 good
**Presentation:** 4 excellent
**Contribution:** 4 excellent
**Rating:** 7
**Confidence:** 4

**Summary:**

This paper proposes a new zeroth order optimizer, MeZO, for LM training.  This is proposed as an improvement to ZO-SGD.  The advantage of this approach is a 12x reduction in the amount of memory required for training compared to backpropagation.  This enables the training of much larger models.

The effectiveness of MeZO is shown across a range of benchmarks and model sizes.  The results compare favorably to linear probing and in context learning.

**Strengths:**

The MeZO technique stands to unlock substantial capability for LM training.  This enables training of much larger networks.  The compatibility with LoRA, prefix tuning are important use cases for many LM users.  There is an ability for optimizing non-differentiable objectives which is compelling, and could be expanded in the future.

The empirical behavior are coupled with a section on theory which effectively describes the both the expected behavior, but elaborates on the expected slow convergence by expanding the theoretical analysis to address low effective rank networks.

**Weaknesses:**

While the analysis refers to the convergence rate of MeZO, there is a very brief treatment of convergence behavior in the paper (Appendix E.2) It might be helpful for this to be expanded and possibly compared to backpropagation, especially in the context of the presentation of Section 4 Theory.

**Questions:**

Appendix A (and Section 3) demonstrate that promoting is crucial to MeZO performance.  Why is this? Much of the other behavior is supported by a theoretical treatment, but this observation stands out as relatively uninterrogated.

**Limitations:**

yes though fairly lightly.  The observation about prompts being critical for training may be a limitation for some (new) tasks or datasets.

---

> ### Author Rebuttal · Authors · 2023-08-10
>
> **Can the theoretical convergence analysis of MeZO be compared to backpropagation?**
>
> Corollary 1 directly compares the SGD convergence rate to the convergence rate of MeZO, since the term in brackets in equation 5 is the per-step loss decrease of SGD (see Lemma 1). Two factors make MeZO converge more slowly than standard backpropagation (lines 211-218): (1) MeZO has to be run with a smaller learning rate than SGD in order to reliably decrease the loss at each step, and (2) MeZO reduces the amount that the loss can decrease at each step.
>
> **Why is prompting crucial to MeZO?**
>
> Please refer to our general response. In short, we hypothesize that using a prompt makes the fine-tuning objective similar to the pre-training one, which likely exhibits a Hessian with low effective rank.

---

> > ### Comment · Reviewer_Qhm1 · 2023-08-18
> >
> > Thank you for your responses

---

### Official Review · Reviewer_unzy · 2023-07-06

**Soundness:** 3 good
**Presentation:** 3 good
**Contribution:** 4 excellent
**Rating:** 7
**Confidence:** 4

**Summary:**

The paper present a new optimiser MeZo based on stochastic approximation using gradient perturbation.
This optimiser is very memory efficient as it only requires to perform 2 forward passes with different deltas/epsilons on the parameters and multiple gaussian samplings.   These algorithms allows "finetunning" large language models to specific tasks very efficiently ( up to 30B on a single A100) yielding between x4 to x12 memory reductions. Since the proposed algorithm is an optimizer it can be combined with other standard techniques such as LORA or prefix tunning. All this is applied in finetunning setups similar to ICL.

**Strengths:**

The algorithm is a new application of well known stochastic gradient approximation but many times forgotten due to their slowness.
It is very surprising that this algorithm works, and the authors provide a theoretical justification of why this could be working in this case.
They also acknowledge that despite of what it might sound this approach only works in prompting fine-tunning scenario [Appendix B.2].

The results are insight full , the baselines are fair and the theoretical analysis is correct to the best of my knowledge.

This can settle as an alternative to in context learning with prompting, by  fine-tuning with  a limited set of examples (512).

The proposed technique seems to work on the benchmarks used and seems to surpass other techniques such as Zero-shot, LP, ICL, and approaches very close performance to fine-tunning. This can be set a cheap alternative to ICL for some tasks.



**Weaknesses:**

While there was a huge and titanic effort in the paper there were many questions that steam from the technique.
The first area which has not been explored too much and given as true is the need of having a prompt to apply MeZO.
This key ingredient is not well studied but rather given on some preliminary experiments, e.g. Table 5.
Why is MeZO not working w/o prompts, even some very simple prompts ?

The need of the prompt also raises the question of how this is related to ICL , as there are some works that suggest ICL maybe doing some alike to fine-tuning or back-propagation though the attention weights. Is this combination of prompting and MeZo that is guiding the forward propagations ? is there a mixed cooperation between the prompt and the stochastic technique ? How much of the prompt is needed ?

Another question would be how the different techniques behave as a function of the number of examples k. It would have been nice to see a plot for some models at least between ICL , MeZo, and possible ft on the selected tasks. Why have authors stopped at 512 ? why only 16 or 512 ? there are some dataset that contain more training examples. Why didn't they compare fine-tuning and MeZo in other setups with larger examples ?

There is the relationship between the task itself and the optimiser. It is not clear to me, in which tasks this will work properly. I suspect given the prompting above that this might only work on low-perplexity tasks or task in which prompting or ICL can generate good results and not in other more complex tasks.

Clearly this is maybe too much to address in the paper, but all aspects above point toward the little understanding the reader is left with on under which conditions this technique can be applied. The future work seems to already assume the MeZO algorithm is working and proved, but there is just an hypotheses and a very low link between the experiment conditions and the theory. The link is stablished as "We attribute these phenomena to the Hessian of the loss exhibiting  small local effective rank." . It would have been nice to strengthen this connection with some experiments or computation. Could this explain when or how this algorithm is applied ? if we remove the prompt does this increases the H effective rank ? would other tasks exhibit larger ranks ?  how can we reduce it for each of the tasks ?


Please, I would kindly ask the authors to read above questions and discussion as a signal of the interest the paper brought to the research field and not as criticism on their very interesting work.


**Questions:**

While I have many question none of them affect the paper directly.
It would be nice however if some of the weakness could be discussed by authors:
* in which task will this technique work? do you have a characterisation ?  have you tried other tasks where it failed ?
* why the Hessian hypotheses ? is this based on some experimental or preliminar analysis ?
* when ft surpasses this techniques? when we have to ft on hundreds of thousands of examples ?
* why it only works with prompting ?


**Limitations:**

While the author do not focus on the limitations,  they are more or less clear per previous analysis. The focus of the paper is more on the direction of stressing the surprise of the technique with all known drawbacks is actually workig.

---

> ### Author Rebuttal · Authors · 2023-08-10
>
> **Why does MeZO require using a prompt? What tasks can MeZO work on? Why do you need the Hessian hypothesis (Assumption 1)?**
>
> Please refer to our general response. In short, we hypothesize that using a prompt makes the fine-tuning objective similar to the pre-training one, which likely has a Hessian with a low effective rank. Following this, we agree with you that a task that has a lower perplexity (i.e., a more “natural” prompt) will probably work better with MeZO. Once we added a simple prompt, we did not encounter any tasks that MeZO completely failed to train on.
>
> **How does the empirical success of MeZO interact with the theoretical hypothesis that transformers may simulate fine-tuning on a smaller, internal model during inference time?**
>
> MeZO is a useful tool for fine-tuning currently popular LLMs trained with standard pre-training practices. It is not guaranteed to work for models that are designed or trained in new ways, such as the scenario you mention. It may be the case that the internal model is not stable to fine-tuning the large model. Alternatively, it could be stable to fine-tuning in some way (e.g., analogous to noise-tolerant circuits) that allows it to not be destroyed during MeZO. Overall, we are not sure if currently existing pre-trained models are simulating and updating internal models, so we cannot be sure how such constructed models would behave during fine-tuning, whether it is done with backpropagation or MeZO.
>
> **How does MeZO behave with different numbers of examples?**
>
> Thanks for your question. We will include experiments ablating against different dataset sizes in a subsequent revision.

---

> > ### Comment · Reviewer_unzy · 2023-08-12
> >
> > Thanks for kindly answering my questions.

---

### Official Review · Reviewer_x4kH · 2023-07-07

**Soundness:** 4 excellent
**Presentation:** 4 excellent
**Contribution:** 4 excellent
**Rating:** 8
**Confidence:** 4

**Summary:**

The paper proposes an enhanced memory efficient zero-order optimization method named MeZO. MeZO only requires the same memory as inference time and thus can enable model tuning for large LMs with limited memory budget. The authors demonstrate the efficacy of MeZO on multiple NLP benchmarks compared with linear probing, in context learning and fine-tuning in few/low shot learning regime. The authors also provided detailed theoretical proof for MeZO.

**Strengths:**

0 - The authors targeted a not well understood domain (zero order optimization) and open up new opportunities for future works. In the era of LLMs, this method can enable many future work especially for those who don't have access to large-scale GPU clusters.
1 - Comprehensive experiments and ablation studies on the proposed method.
2 - Strong theoretical support on the proposed method.
3 - Good writing and flow which makes the paper easy to follow and understand.
4 - Well-articulated future work.

**Weaknesses:**

No major weaknesses. I left some comments in the questions section and hope  the authors can answer and address.

**Questions:**

- Can authors also report practical training time compared with standard fine-tuning (e.g., in terms of # steps/ second)?

- Maybe report average numbers as well in experiment results (e.g., Table 1)

- Interested to see how the performance of MeZO compares with fine-tuning in the scenario of full training data (rather than k = 100, 500, etc).

**Limitations:**

The authors have discussed limitations in the conclusion section and aim to explore them in future work.

---

> ### Author Rebuttal · Authors · 2023-08-10
>
> Thank you for your suggestion, and we will report average numbers in the experiment results in the next revision.
>
> **What is the practical training time compared to standard fine-tuning?**
>
> Please refer to our general response for a wall clock time analysis. In short, MeZO reduces the number of GPU-hours needed to train large models, leading to a 2x GPU-hour reduction on a 30B model compared to Adam fine-tuning.
>
> **How does MeZO perform with full training data?**
>
> We choose the fixed number of training example setting because of compute limitations. Some datasets have millions of examples, so fine-tuning on the entire dataset can be very expensive for the model scale we are studying. We will include more ablations showing how MeZO performance changes with the dataset size in the next revision.

---

> > ### Comment · Reviewer_x4kH · 2023-08-16
> >
> > Thank you.

---

### Official Review · Reviewer_qocq · 2023-07-07

**Soundness:** 4 excellent
**Presentation:** 4 excellent
**Contribution:** 3 good
**Rating:** 8
**Confidence:** 4

**Summary:**

This work introduced a memory-efficient zeroth-order optimizer that can fine-tune large language models with the same memory footprint as inference, using only forward passes and gradient estimates. Comprehensive experiments across model types, scales, and tasks, showing that MeZO outperforms zero-shot, in-context learning, and linear probing, and achieves comparable performance to fine-tuning with backpropagation, while reducing memory cost by up to 12 times. Non-differentiable objectives that MeZO can optimize, such as accuracy or F1 score, which are usually not amenable to backpropagation. Theoretical insights that explain why MeZO can optimize LMs with billions of parameters, despite classical zeroth-order analyses suggesting otherwise.


**Strengths:**

1. It proposes a novel and memory-efficient method to fine-tune large language models without backpropagation, which can save up to 12x memory compared to standard methods.

2. It demonstrates that the proposed method can achieve comparable or superior performance to fine-tuning with backpropagation across various tasks, models, and tuning techniques.

3. It shows that the proposed method can optimize non-differentiable objectives, such as accuracy or F1 score, which are useful for many applications.

4. It provides theoretical insights on why the proposed method can overcome the classical limitations of zeroth-order optimization and leverage the benefits of pre-training and task prompts.


**Weaknesses:**

1. While the experiments demonstrate good performance on the language understanding tasks, it is unknown whether the method is also applicable to the generation tasks.

2. It relies on the assumption of low effective rank of the Hessian matrix, which may not hold for all loss functions. It would be great to have a discussion about the scope of application for the proposed method.


**Questions:**

How is the training stability of MeZO? Does it an advantage over the backpropagation-based methods, especially for the large models?

**Limitations:**

1. It would be better to have the experiment results on the generation tasks, e.g. translation and summarization.

2. I suggest the authors to have a discussion about the scope of application for the proposed method.

---

> ### Author Rebuttal · Authors · 2023-08-10
>
> **Does MeZO work for generation tasks?**
>
> Table 1 and Figure 1 show the performance of MeZO on DROP and SQuAD, which are two question-answering tasks that are formatted as generation tasks in our experiments. For each task, given the question, we train the model to directly generate the answer text (please see our Table 12 for details). We leave the study of more generation tasks like summarization and translation to future work.
>
> **When does the assumption of the low effective rank of the Hessian hold?**
>
> Please refer to our general response. We hypothesize that when using a good prompt, the Hessian of the downstream objective likely exhibits a low effective rank.
>
> **How stable is MeZO training? Does it have an advantage over backpropagation?**
>
> MeZO is not very sensitive to hyperparameter choices. As shown in Tables 13 (RoBERTa-large) and 14 (OPT), we restrict the grid searches to a very narrow range of hyperparameters and often test MeZO with fewer configurations than we use to test fine-tuning with backpropagation. Also, as a gradient estimate, MeZO avoids well-known issues with backpropagation such as vanishing and exploding gradients, though these rarely occur when training networks with residual connections. MeZO is unstable in other ways, like if $\epsilon$ must be set very small. However, in practice, we find that MeZO succeeds with a relatively large $\epsilon$ and reduces the loss consistently over the course of training.

---

> > ### Comment · Reviewer_qocq · 2023-08-15
> >
> > Thank you for answering my question.

---

### Author Rebuttal · Authors · 2023-08-10

We thank all reviewers for their valuable feedback. We address some shared questions here.

**When can MeZO succeed in fine-tuning? What losses satisfy Assumption 1 (i.e., the Hessian has a low effective rank)? Why is a prompt necessary for MeZO to be able to fine-tune the model? Can you verify the dependence of MeZO convergence rate on the effective rank?**

Our theory in Section 4 provides a sufficient (but not necessary) condition for MeZO to succeed: the Hessian should exhibit a small local effective rank during fine-tuning (Assumption 1). We hypothesize that the Hessian of the pre-training objective likely exhibits low effective rank, because the model has been trained for many steps during pre-training. Ample evidence suggests that training for many steps can make the Hessian have a low effective rank in the case of vision (see lines 222-228). Adding a prompt turns the downstream task into next-word prediction [1] (or masked language modeling [2]). So, the Hessian of the fine-tuning objective when using a prompt (similar to pre-training) likely exhibits a small effective rank like the pre-training one [1, 3]. There is additional empirical evidence that the Hessian of a language model during fine-tuning likely has low rank [4].

Additionally, per reviewer XvSx’s suggestion, we ran simulations in a simple setting to verify the dependence of MeZO convergence rate on the effective rank and reported the results in the attached PDF. We observed that the slowdown of the convergence scales with the effective rank. We will include these experiments in the next revision of the paper.

[1] Nikunj Saunshi, Sadhika Malladi, Sanjeev Arora. A Mathematical Exploration of Why Language Models Help Solve Downstream Tasks. ICLR 2021.

[2] Tianyu Gao, Adam Fisch, Danqi Chen. Making Pre-trained Language Models Better Few-shot Learners. ACL 2021.

[3] Sadhika Malladi, Alexander Wettig, Dingli Yu, Danqi Chen, Sanjeev Arora. A Kernel-Based View of Language Model Fine-Tuning. ICML 2023.

[4] Armen Aghajanyan, Sonal Gupta, and Luke Zettlemoyer. Intrinsic Dimensionality Explains the Effectiveness of Language Model Fine-Tuning. ACL 2021.

**What is the wall-clock efficiency of MeZO compared to standard training with backpropagation?**

The attached PDF with this rebuttal shows the wall-clock efficiency of MeZO compared to fine-tuning. MeZO takes more steps to achieve similar performance than fine-tuning, but requires much less wall-clock time per step and requires fewer GPUs. The gains are more prominent for larger models, which are the ones that require more memory to fine-tune. For example, for a 30B model, we show that MeZO enjoys a 7.74x per-step speed up and a 2x total GPU-hour reduction compared to fine-tuning with Adam. We will include these results in the next revision of the paper.

---

### Decision · Program_Chairs · 2023-09-21

**Decision:**

Accept (oral)

**Comment:**

This paper proposed a memory-efficient zeroth-order optimizer that can fine-tune large language models using only forward passes. All reviewers give positive ratings to this paper and mention some merits of this paper such as targeting a not well-understood domain (zero order optimization), with good experiment and theoretical support, and good paper writing. Although reviewers ask several questions including whether it can apply to the generation task, or when the assumption of the low effective rank of the Hessian matrix holds, the authors answer these questions well. Overall, it is a strong paper with a clear problem setting, nice experiment results, theoretical support, and practical application. I recommend accepting this paper as a spotlight.